# Integrated Metabolomics and Transcriptomics Analyses Reveal the Critical Role of Caffeic Acid in Potato (*Solanum tuberosum* L.) Cold Tolerance

**DOI:** 10.3390/plants14223447

**Published:** 2025-11-11

**Authors:** Xiang Li, Guonan Fang, Yongzhen Ma, Wang Su, Shenglong Yang, Yun Zhou, Yanping Zhang, Jian Wang

**Affiliations:** 1Academy of Agriculture and Forestry Sciences, Qinghai University, Xining 810000, China; 15500061802@163.com (X.L.); 17755203950@163.com (G.F.); 2025990004@qhu.edu.cn (Y.M.); suwangtt@126.com (W.S.); ysl890224@163.com (S.Y.); zhouyun75@163.com (Y.Z.); 2Qinghai Provincial Key Laboratory of Potato Breeding, Qinghai University, Xining 810000, China; 3Ministry of Education Engineering Research Center of Potato in Northwest Region, Qinghai University, Xining 810000, China

**Keywords:** potato (*Solanum tuberosum* L.), cold stress, caffeic acid, differentially accumulated metabolites, differentially expressed genes, KEGG pathway

## Abstract

Owing to the high altitude and short frost-free period of the Tibetan Plateau, potato plants are frequently exposed to cold stress (CS), which severely restricts their growth and productivity. Thus, understanding the mechanisms underlying cold tolerance in potato varieties is crucial for breeding improvement. This study aims to investigate the role of caffeic acid in potato cold tolerance and to elucidate the molecular mechanisms underlying the CS response. To achieve this, we conducted comprehensive metabolomic and transcriptomic analyses of KY130 (cold-tolerant) and KY140 (cold-sensitive) potato cultivars under CS at the seedling stage. ELISA results showed that caffeic acid levels were higher in KY130 than in KY140, while CS-KY130 exhibited higher levels than those of CS-KY140. Across all treatments, KY130 showed significantly higher activities of antioxidant enzymes (CAT and SOD) and higher contents of osmolytes (proline, soluble protein, and soluble sugar) than those of KY140. Caffeic acid and naringenin* were identified as candidate metabolites potentially involved in the direct and indirect cold resistance of potatoes. *StPAL*(*Soltu.Atl.03_2G004060*, *Soltu.Atl.03_2G004070*, *Soltu.Atl.03_2G008130*) and *StCSE*(*Soltu.Atl.04_1G006370* and *Soltu.Atl.04_3G005440*), identified as upstream regulators of caffeic acid, were associated with the direct and indirect cold resistance of potatoes. KEGG pathway analysis of differentially accumulated metabolites and differentially expressed genes revealed several key metabolic pathways, including “flavonoid-related metabolism,” “lipid metabolism,” and “amino acid metabolism.” This research enhances our understanding of caffeic acid and the molecular mechanisms involved in the response of potatoes to CS, and supports future efforts in potato screening and breeding programs.

## 1. Introduction

The potato (*Solanum tuberosum* L.) is an annual herbaceous plant belonging to the *Solanaceae* family and the genus *Solanum*. It serves as a nutritionally dense crop rich in diverse bioactive constituents, including dry matter, starch, proteins, fats, vitamins (such as vitamin C, niacin, pantothenic acid, and folic acid, among others), reducing sugars, minerals (including calcium, potassium, and zinc, among others), anthocyanins, energy, and carbohydrates among others [1]. Potato is the fourth most important food crop in China, contributing significantly to national food security, enhancing dietary nutrition, and promoting regional economic development [2]. While potatoes thrive in temperate and cool environments, they exhibit limited tolerance to freezing and low temperatures [3]. Cold stress (CS) is therefore a major environmental factor limiting the growth and development of potato plants [2]. It primarily inhibits bud germination and reduces seedling emergence. During the seedling stage, CS impairs photosynthetic efficiency and slows vegetative growth, ultimately reducing yield. At later developmental stages, CS can cause premature senescence and plant death before full maturity, further diminishing productivity. Moreover, exposure to CS during harvest can damage tubers, increasing their susceptibility to pathogenic infection, decay, and postharvest storage losses [2]. Through evolutionary adaptation, plants have developed multifaceted strategies to tolerate low-temperature stress, involving coordinated morphological, physiological, biochemical, and molecular responses, including gene regulation, enzyme activity modulation, and metabolic stabilization [4,5,6,7]. Elucidating the molecular mechanisms underlying potato responses to low-temperature stress provides a theoretical foundation and technical framework for developing cold-tolerant cultivars.

Studies show that prolonged exposure to low-temperature stress can induce cold acclimation in some plants through osmotic regulation and enzyme activity [8]. Osmotic regulatory substances can reduce water potential, limit water loss, and enhance cold tolerance, while antioxidant enzymes mitigate oxidative damage by scavenging reactive oxygen species generated under CS [9,10,11].

Omics-based approaches are transforming our understanding of cold adaptation in plants [4,12]. As a key discipline within systems biology, metabolomics focuses on identifying and quantifying metabolites within a cell or organism. Metabolites represent direct functional links between genes and phenotypes, offering valuable insights into the underlying regulatory mechanisms [13,14]. Metabolomics has become a powerful analytical platform for elucidating plant responses to CS [4,15,16,17]. In addition to metabolomics, transcriptomics is a key approach for elucidating plant responses to CS [18,19,20]. However, the regulatory mechanisms underlying plant stress responses are highly complex, and single-omics analyses often provide only a partial view of the intricate regulatory networks and crosstalk involved in CS responses. Integrating tightly coordinated metabolic and sophisticated gene regulatory networks constitutes a key adaptive mechanism in plants under CS [12,21,22,23]. Single- and multi-omics analyses of potatoes under CS reveal the involvement of lipids, flavonoids, amino acids, and carbohydrates [24], along with lipid metabolism [25], linoleic acid metabolism [24], and the arginine decarboxylase gene *ADC1*, which functions in the putrescine pathway [26] and plays an important role in stress adaptation. Current research on potato cold tolerance remains limited, and the underlying mechanisms are not yet fully understood. Therefore, comprehensive and integrative studies are required to elucidate this complex adaptive process.

Caffeic acid—known for its antioxidant and free radical-scavenging properties [27]—and cinnamic acid may enhance cold tolerance in wheat [28] and potato [29]. Enzyme-linked immunosorbent assay (ELISA) findings reveal that caffeic acid plays an important role, while cinnamic acid is not a major contributor to the cold resistance of KY130. Therefore, this study aims to investigate the cold resistance mechanisms associated with caffeic acid in potato through metabolomic and transcriptomic analyses of leaf samples from KY130 and KY140 under CS. The study findings could enhance our understanding of caffeic acid and the molecular mechanisms underlying potato response to CS, while further extending and validating previous findings [29].

## 2. Materials and Methods

### 2.1. Plant Growth Conditions

Virus-free tissue culture seedlings of KY130 (cold-tolerant) and KY140 (cold-sensitive) potato cultivars were grown on Murashige and Skoog medium solidified with carrageenan (Shijiazhuang Nutrition Tissue Culture Technology Co., Ltd., Shijiazhuang, China) for 18 days. The culture room was maintained at an illumination intensity of 2500 lx under a 16 h light/8 h dark photoperiod. A low-temperature incubator (Shanghai YiHeng Scientific Instruments Co., Ltd., Shanghai, China) was used for the experiment.

### 2.2. Experimental Design for Phenotyping with Statistical Evaluation

All samples were divided into control (CK, 22–25 °C) and CS (−4 °C) treatment groups for 7 days, followed by a 12–24 h recovery period at 22–25 °C before phenotypic evaluation. CS significantly affected plant growth, as evidenced by leaf damage. Leaves with a wilted surface area of ≥ 50% were classified as damaged. The leaf damage rate under CS was calculated as the ratio of damaged leaves to the total number of leaves per plant. KY130 and KY140 exhibited different degrees of leaf damage [29]. Mean leaf damage rates for KY130 and KY140 cultivars were determined from 60 cold-stressed plants per genotype.

### 2.3. Measurement of Caffeic Acid and Cinnamic Acid Levels Using Enzyme-Linked Immunosorbent Assay

Caffeic and cinnamic acid contents were measured in the control (CK, maintained at 22–25 °C) and treatment groups (exposed to −4 °C for 12 h). Caffeic and cinnamic acid levels in potato samples were quantified using commercial ELISA kits (Jiangsu Meimian Industrial Co., Ltd., Yancheng, China) following the instructions of the manufacturer. All analyses were performed in triplicate.

### 2.4. Physiological Indicators Analysis

Physiological parameters were assessed in the control (CK, maintained at 22–25 °C) and treatment groups (exposed to −4 °C for 6, 12, and 18 h). Enzymatic activities (CAT and SOD) and osmolyte content (proline, soluble protein, and soluble sugar) were quantified using commercial assay kits (Nanjing Jiancheng Bioengineering Institute, Nanjing, China) with an ultraviolet–visible spectrophotometer (Unico (Shanghai) Instrument Co., Ltd., Shanghai, China), following the instructions of the manufacturer. Each treatment included at least six biological replicates.

### 2.5. Metabolomic and Transcriptomic Analysis

#### 2.5.1. Sample Preparation for Metabolomic and Transcriptomic Analysis

Comparative multi-omics analyses (metabolomic and transcriptomic) were conducted using samples from the control group (CK, 22–25 °C) and the 12 h cold-treated group (−4 °C), followed by quantitative real-time polymerase chain reaction (qRT-PCR) analysis. All experiments included three independent biological replicates to ensure statistical reliability.

#### 2.5.2. Metabolomic Analysis

Metabolite identification was conducted using the MetWare Database (MWDB; MetWare Biotechnology Co., Ltd., Wuhan, China). The sample extraction procedure involved vacuum freeze-drying, grinding into a fine powder, methanol extraction, and filtration of the supernatant through a 0.22 μm microporous membrane for subsequent analysis. Chromatographic and mass spectrometric analyses were conducted using an ultra-performance liquid chromatography system [ExionLC™ AD; SCIEX, https://sciex.com.cn/ (accessed on 3 August 2022)] coupled with a tandem mass spectrometer [Applied Biosystems 6500 QTRAP; SCIEX, https://sciex.com.cn/ (accessed on 3 August 2022)]. Metabolite characterization was conducted using the self-built MWDB. Metabolite quantification was conducted in multiple reaction monitoring mode on a triple quadrupole mass spectrometer. Differentially accumulated metabolites (DAMs) were identified based on the following criteria: variable importance in projection ≥ 1 and fold change (FC) ≥ 2 or FC ≤ 0.5. DAMs were defined as metabolites showing significant differences between comparison groups. Screening criteria for DAMs potentially associated with potato cold resistance were established as follows: ① Direct cold resistance: upregulated DAMs in KY140 vs. KY130; and ② Indirect cold resistance: upregulated DAMs in CS-KY140 vs. CS-KY130.

#### 2.5.3. Transcriptomic Analysis

cDNA libraries were sequenced on the Illumina sequencing platform by MetWare Biotechnology Co., Ltd. (Wuhan, China). Sample collection and preparation included RNA quantification and qualification, library construction for transcriptome sequencing, and subsequent clustering and sequencing. Differentially expressed genes (DEGs) were identified using the following criteria: |log_2_FC| ≥ 1 and false discovery rate < 0.05. Screening criteria for DEGs potentially associated with potato cold resistance were defined as follows: ① Direct cold resistance: upregulated DEGs in KY140 vs. KY130; and ② Indirect cold resistance: upregulated DEGs in CS-KY140 vs. CS-KY130.

### 2.6. Quantitative Reverse Transcription Polymerase Chain Reaction Analysis

Total RNA was extracted from leaf samples using the RNAprep Pure Polysaccharide and Polyphenol Plant Total RNA Extraction Kit (Tiangen Biotech Co., Ltd., Beijing, China). cDNA was then synthesized through reverse transcription with the PrimeScript™ RT Master Mix (Perfect Real Time; TaKaRa Biotechnology Co., Ltd., Dalian, China). qRT-PCR was conducted using TB Green^®^ Premix Ex Taq™ II (Tli RNaseH Plus; TaKaRa Biotechnology Co., Ltd., Dalian, China) on a Roche LightCycler^®^ 96 qRT-PCR System (Basel, Switzerland). The qRT-PCR results were analyzed using the 2^−∆∆Ct^ method (Expression level = [2,-Cq1/2,-Cq2] × 100, where Cq1 = target gene and Cq2 = actin from the same sample; actin was set to 100; Table 1 and Table 2).

### 2.7. Data Analysis

Phenotypic data, physiological indices, and qRT-PCR results were analyzed using the data summary functions in Microsoft Office Excel 2019. GraphPad Prism 8 was employed to generate bar graphs and perform statistical analysis. Statistical significance was assessed through one-way analysis of variance (ANOVA) and Tukey’s multiple comparisons test.

## 3. Results

### 3.1. Cold Stress-Induced Phenotypic Variation in Potatoes

KY140 exhibited greater cold-induced leaf damage (57.1%) than that of KY130 (32.2%) after 7 days of CS and 12–24 h recovery at 22–25 °C (Figure 1A,B).

### 3.2. Alterations in Small Molecule Indicator and Physiological Parameters in Potatoes Under Cold Stress

ELISA analysis revealed that caffeic acid levels were higher in the treatment groups (CS-KY140 and CS-KY130) than in the corresponding control groups (KY140 and KY130). Furthermore, the control group KY130 had higher caffeic acid content than that of the control KY140, while the CS-KY130 group showed higher levels than those of CS-KY140 (Figure 2A). These findings were consistent with the metabolomic data. Cinnamic acid levels in the treatment groups (CS-KY140 and CS-KY130) were higher than those in the corresponding control groups (KY140 and KY130) (Figure 2B), aligning with the metabolomic trends. Across all treatments, KY130 showed significantly higher antioxidant enzyme activities (CAT and SOD) and higher osmolyte contents (proline, soluble protein, and soluble sugar) than those of KY140 (Figure 2C–Q).

### 3.3. Metabolome Profiling of Potatoes Exposed to Cold Stress

Principal component analysis (PCA) with confidence intervals clearly demonstrated sample clustering s within groups and separation trends between groups. Metabolomic data quality was validated through PCA, confirming that it met all requirements for further investigation (Figure 3A). Loading analysis revealed metabolites contributing to the separation of principal components (Figure 3B). Comparative analysis revealed distinct differential accumulation patterns among the four comparison groups: In total, 195 DAMs were identified in KY140 vs. CS-KY140, 258 in KY130 vs. CS-KY130, and 124 DAMs were shared between these two comparisons. In KY140 vs. KY130, 571 DAMs were identified, while 587 DAMs were detected in CS-KY140 vs. CS-KY130, with 427 common to both comparisons (Figure 3C). Comparative analysis revealed distinct DAM accumulation patterns. In KY140 vs. CS-KY140, 161 DAMs were upregulated and 34 were downregulated, whereas in KY130 vs. CS-KY130, 228 DAMs were upregulated and 30 were downregulated. Genotypic comparisons demonstrated that 392 DAMs were upregulated and 179 were downregulated in KY140 vs. KY130, while 421 were upregulated and 166 were downregulated in CS-KY140 vs. CS-KY130. Our analysis revealed 519 DAMs potentially associated with potato cold resistance, including 392 DAMs that may be linked to direct cold resistance and 421 that may be related to indirect cold resistance (Figure 3D). These 519 DAMs comprised various metabolite classes: lipids (82), flavonoids (122), terpenoids (16), phenolic acids (100), organic acids (18), alkaloids (82), lignans and coumarins (21), amino acids and derivatives (30), nucleotides and derivatives (18), and other compounds (30). Compared with KY140, KY130 exhibited a higher number of DAMs with elevated levels, and a similar pattern was observed between CS-KY130 and CS-KY140 (Figure 3E). These findings indicate distinct DAM expression profiles between the two cultivars under CS treatment.

### 3.4. Transcriptomic Analysis of Potatoes in Response to Cold Stress

Correlation analysis of the transcriptomic data confirmed that the dataset met the quality criteria for subsequent analyses (Figure 4A). In total, 40,082 DEGs were identified across the four comparisons (Figure 4B). The KY140 vs. CS-KY140 comparison resulted in more unique DEGs than those of KY130 vs. CS-KY130. Conversely, fewer unique DEGs were detected in KY140 vs. KY130 relative to CS-KY140 vs. CS-KY130 (Figure 4B). Upregulated and downregulated DEGs were more abundant in KY140 vs. CS-KY140 than in KY130 vs. CS-KY130. Additionally, KY140 vs. KY130 exhibited more upregulated but fewer downregulated DEGs than those of CS-KY140 vs. CS-KY130 (Figure 4C). We identified 16,589 and 13,974 DEGs potentially associated with direct and indirect cold resistance of potatoes, respectively (Figure 4C). These findings suggest that the two cultivars exhibit different DEG expression patterns in response to CS.

### 3.5. Identification and Expression Analysis of Transcription Factors Involved in Cold Stress Response

Transcription factors (TFs) play a crucial role in regulating plant responses to CS. Across the comparisons KY130 vs. CS-KY130, KY140 vs. CS-KY140, KY140 vs. KY130, and CS-KY140 vs. CS-KY130, 2624 TFs were detected, primarily belonging to 72 TF families, including MYB (175), AP2/ERF (165), bHLH (148), HB (137), C2C2 (131), WRKY (121), and others (Figure 5A). Among them, 286 TFs and 239 TFs were detected in KY140 vs. CS-KY140 and KY130 vs. CS-KY130, respectively, with 97 TFs common to both comparisons. Furthermore, 1756 TFs and 1983 TFs were identified in KY140 vs. KY130 and CS-KY140 vs. CS-KY130, respectively, including 1248 shared TFs (Figure 5B). In KY140 vs. CS-KY140, 258 upregulated and 28 downregulated TFs were detected, while 214 upregulated and 25 downregulated TFs were detected in KY130 vs. CS-KY130. In KY140 vs. KY130, 1002 upregulated and 754 downregulated TFs were detected, while 882 upregulated and 1101 downregulated TFs were identified in CS-KY140 vs. CS-KY130 (Figure 5C). In total, 1002 TFs belonging to 68 TF families—including MYB (63), HB (60), bHLH (54), C2C2 (48), WRKY (48), AP2/ERF (46), and others—may be related to the direct cold resistance of potatoes (Figure 5D). Overall, 882 TFs belonging to 66 TF families, including MYB (55), bHLH (50), WRKY (50), C2C2 (45), HB (44), AP2/ERF (42), and others, may be associated with the indirect cold resistance of potatoes (Figure 5E). These findings reveal distinct TF expression patterns between the two cultivars under CS conditions.

### 3.6. RNA-Seq Data Validation via Quantitative Reverse Transcription Polymerase Chain Reaction

qRT-PCR was performed to validate the expression profiles of eight cold-tolerance-related DEGs in both potato cultivars under CS. The strong consistency between the qRT-PCR and RNA-Seq results confirmed the reliability and accuracy of the transcriptomic data (Figure 6).

### 3.7. Gene Ontology Enrichment Analysis

Gene Ontology (GO) enrichment analysis was conducted to elucidate the functional roles and biological processes associated with the DEGs in KY130 vs. CS-KY130 and KY140 vs. CS-KY140. Among the 4002 DEGs, 3379 were annotated to 41 GO categories, including “biological process,” “molecular function,” and “cellular component.” Most DEGs in the “biological process” category were enriched in “cellular process” and “metabolic process.” Within the “molecular function,” “binding” and “catalytic activity” represented the majority of DEGs annotations. In the “cellular component,” DEGs were exclusively classified under “cellular anatomical entity” and “protein-containing complex.” These findings suggest that these pathways are significant involved in the response of the potato to CS (Figure 7).

### 3.8. Integrated Metabolomic and Transcriptomic Analysis to Reveal Crucial Kyoto Encyclopedia of Genes and Genomes Pathways Responsive to Cold Stress

The metabolomic and transcriptomic datasets were integrated to further reveal the responses of KY130 and KY140 to CS. Gene–metabolite network analysis facilitates understanding of functional relationships and discovery of previously unidentified regulatory components. In the comparison between KY130 and CS-KY130, DAMs were enriched across 58 KEGG pathways, while DEGs were enriched across 122 KEGG pathways, with 54 pathways showing common enrichment via DAMs and DEGs. Similarly, in KY140 vs. CS-KY140, DAMs were enriched across 31 KEGG pathways, while DEGs were enriched across 129 KEGG pathways, with 30 pathways showing co-enrichment. KEGG pathway analysis of DAMs and DEGs revealed significant enrichment across several metabolic categories, particularly flavonoid-related, along with core metabolic pathways involving lipids, amino acids, carbohydrates, nucleotides, and energy (Figure 8).

### 3.9. “Flavonoid-Related Metabolism” in Response to Cold Stress

Caffeic acid, naringenin*, butin, (+)-afzelechin, 5-O-p-coumaroylquinic acid*, dihydroquercetin, dihydrokaempferol, and myricetin are likely to contribute directly and indirectly to potato cold resistance. Sinapoyl alcohol and kaempferol 3-sophorotrioside may be associated with the direct cold resistance of potatoes. Conversely, (+)-catechin, cosmetin*, and eriodictyol may be involved in the indirect cold resistance observed in potatoes. Sinapoyl alcohol and kaempferol 3-sophorotrioside showed no significant differences, while butin levels were elevated in KY130 vs. CS-KY130, undetectable in KY140, and detectable in CS-KY140. (+)-Afzelechin, (+)-catechin, and cosmetin* exhibited no significant differences in KY130 vs. CS-KY130; however, these compounds were detected in KY140 but absent in CS-KY140. Dihydrokaempferol levels were elevated in KY130 vs. CS-KY130 were undetectable in KY140 and CS-KY140. DEGs expressing PAL, 4CL, HCT, CAD, COMT, and CYP75B1 exhibited increased expression in KY130 vs. CS-KY130 and KY140 vs. CS-KY140. Among these, *StCYP73A*(*Soltu.Atl.06_4G021440*), *StCYP75B1*(*Soltu.Atl.08_1G014450* and *Soltu.Atl.08_2G016980*), *StHCT*(*Soltu.Atl.01_1G026080*), and *StPAL*(*Soltu.Atl.06_2G003190*, *Soltu.Atl.09_3G003640* and *Soltu.Atl.10_2G008250*) may be associated with the direct cold resistance of potato. *StCAD*(*Soltu.Atl.03_2G008390*), *StCOMT*(*Soltu.Atl.10_4G005750* and *Soltu.Atl.10_4G005760*), and *StHCT*(*Soltu.Atl.01_3G027900*) may contribute to the indirect cold resistance of potato. Additionally, *St4CL*(*Soltu.Atl.03_2G024490* and *Soltu.Atl.11_3G016410*), *StCOMT*(*Soltu.Atl.10_4G005740*), *StCSE*(*Soltu.Atl.04_1G006370* and *Soltu.Atl.04_3G005440*), *StCYP73A*(*Soltu.Atl.06_4G021450*), *StFLS*(*Soltu.Atl.06_3G012680*), *StHCT*(*Soltu.Atl.S021380*), and *StPAL*(*Soltu.Atl.03_2G004060*, *Soltu.Atl.03_2G004070*, *Soltu.Atl.03_2G008130*) may function in direct and indirect mechanisms of cold resistance of potato (Figure 9).

### 3.10. “Lipid Metabolism” in Response to Cold Stress

The metabolites 13(S)-HODE, 9(S)-HODE, and 13(S)-HOTrE may be related to the direct and indirect cold resistance of potato. The compounds 5,6-DHET, 12(13)-EpOME, 9(S)-HPODE, 2(R)-HOTrE, 9-hydroxy-12-oxo-15(Z)-octadecenoic acid, and 9-hydroxy-12-oxo-10(E),15(Z)-octadecadienoic acid may be associated with the indirect cold resistance of potatoes. The levels of UDP-glucose, traumatic acid, and eicosenoic acid were reduced in KY130 vs. CS-KY130. Traumatic acid and eicosenoic acid were detected in KY130 but were undetectable in CS-KY130. No significant differences were observed among these three DAMs in KY140 vs. CS-KY140. The levels of 5,6-DHET, 12(13)-EpOME, 13(S)-HODE, 9(S)-HPODE, 9(S)-HODE, 9-oxoODE, and 9-hydroxy-12-oxo-10(E),15(Z)-octadecadienoic acid were increased in KY130 vs. CS-KY130, while no significant differences were observed in KY140 vs. CS-KY140. DEGs expressing LOX2S were significantly higher in KY130 vs. CS-KY130 and in KY140 vs. CS-KY140. *StLOX2S*(*Soltu.Atl.03_1G030600*) may play a direct role in conferring cold resistance in potatoes, while *StCEQORH*(*Soltu.Atl.09_1G004340*) may be associated with the indirect cold resistance (Figure 10).

### 3.11. “Amino Acid Metabolism” in Response to Cold Stress

In this study, “amino acid metabolism” primarily encompasses the metabolic pathways of arginine and proline, phenylalanine, and tryptophan. L-Cystine may contribute to the direct and indirect cold resistance of potato, whereas L-homocystine may be associated with the direct cold resistance. 2-Hydroxycinnamic acid and 2-aminophenol may be related to the indirect cold resistance of potatoes. The concentrations of 2-oxoglutarate and fumarate were reduced in KY130 vs. CS-KY130, while the levels of 2-phenylethanol, 2-aminophenol, S-adenosyl-L-homocysteine, α-hydroxycinnamic acid*, 2-hydroxycinnamic acid, 3-hydroxycinnamic acid* and 2-(formylamino)-benzoic acid were increased. No significant differences were observed in these DAMs in KY140 vs. CS-KY140. L-Cystine exhibited no significant variation in KY130 vs. CS-KY130 but showed increased levels in KY140 vs. CS-KY140. Fumarate was detected in KY130 but was undetectable in CS-KY130, while no significant differences were observed in KY140 vs. CS-KY140. α-Hydroxycinnamic acid* and 3-hydroxycinnamic acid* were not detected in KY130 but were detected in CS-KY130. No significant differences were observed in KY140 vs. CS-KY140. L-Homocystine levels were increased in KY130 vs. CS-KY130 and KY140 vs. CS-KY140. 5-Hydroxyindoleacetate was undetectable in KY130 before or after CS treatment. Neither of these two DAMs was detected in KY140, while both were detected in CS-KY140. *StASL*(*Soltu.Atl.04_3G014900*) may be associated with the direct cold resistance of potatoes. *StMIF*(*Soltu.Atl.10_1G007320*) may be involved in the direct and indirect cold resistance of potatoes (Figure 11).

## 4. Discussion

CS is a major abiotic factor that restricts potato growth and development [2]. Potatoes have developed regulatory mechanisms that adjust gene expression and metabolism in response to CS [24,25,26]. Varieties differ in their ability to maintain growth and yield under CS, making omics approaches vital for identifying CS-responsive genes and metabolites [12,24,25,26,30]. In this study, integrated metabolomic and transcriptomic analyses of the KY130 and KY140 potato varieties under CS provide insights into key metabolites (e.g., caffeic acid), genes, and pathways involved in cold tolerance.

### 4.1. Role of Osmotic Regulating Substances and Antioxidant Enzyme Systems in Potato Cold Resistance

Osmotic regulators and antioxidant enzymes are involved in potato low-temperature responses [31]. In this study, KY130 showed significantly higher antioxidant enzyme activities and osmolyte levels than those of KY140, suggesting greater physiological resistance to cold temperatures. Osmotic regulators mitigate cold-induced dehydration stress, while antioxidant enzymes reduce reactive oxygen species accumulation in plant cells under CS [2]. Studies report similar findings in other potato varieties [10,11,32], indicating that osmotic regulators and antioxidant enzyme activities are key indicators of potato cold resistance.

### 4.2. Functional Implications of Differentially Accumulated Metabolites in Potatoes Under Cold Stress

Metabolomic changes more comprehensively reflect physiological alterations in potatoes under different conditions. In this study, the main DAMs involved in the potato low-temperature response include flavonoids, lipids, phenolic acids, and carbohydrates, consistent with previous findings [29]. Catechin [33], eriodictyol [33], and dihydroquercetin [34], and other flavonoids [35,36,37,38,39] exhibit antioxidant activity. Additionally, flavonoid-related compounds respond to low temperatures in freezing-tolerant kiwifruit (*Actinidia arguta*) [40] and cold-tolerant peach (*Prunus persica* L. Batsch) [41]. Integrated metabolomic and transcriptomic analyses revealed higher naringenin accumulation in “Donghe No. 1” (cold-tolerant) than in the “21st Century” (cold-sensitive) peach under CS [41]. ELISA revealed higher caffeic acid levels in the treatment groups (CS-KY140 and CS-KY130) than in their respective controls (KY140 and KY130). Caffeic acid was activated in response to low temperatures under cold conditions. Furthermore, the control KY130 group had higher caffeic acid content than that of the control group KY140, while the CS-KY130 group showed higher levels than those of CS-KY140. These findings suggest that caffeic acid enhances cold tolerance in potatoes. Caffeic acid, a phenolic acid, exhibits antioxidant and free radical-scavenging activity due to its benzene ring containing an ortho-diphenol hydroxyl group [27]. Cinnamic acid levels were also higher in the treatment groups (CS-KY140 and CS-KY130) than in their respective controls (KY140 and KY130). When exposed to CS, cinnamic acid becomes activated in potatoes. These findings align with the ELISA and metabolomic data, suggesting that caffeic acid plays a major role, while cinnamic acid makes a lesser contribution to the cold resistance of KY130. In maize (*Zea mays* L.), metabolomic analysis of two inbred lines––B144 (tolerant) and Q319 (susceptible)––reveals sinapoyl alcohol as a key metabolite associated with cold tolerance [42]. Ferulic acid*, a phenolic compound, exhibits antioxidant activity [43,44], while 2-phenylethanol aids in plant preservation [45]. Integrated metabolomic and transcriptomic analyses of peaches under CS reveal higher accumulations of caffeic and ferulic acids in “Donghe No. 1” (cold-tolerant) than in “21st Century” (cold-sensitive) [41]. Wheat shows elevated cinnamic and caffeic acid levels but reduced naringenin and afzelechin under CS [28]. Phenolic acids and flavonoids contribute to cold tolerance in the resistant variety “KY130” by scavenging free radicals and protecting cells from damage. Biomembranes, which form protective barriers around cells and organelles, are highly sensitive to temperature fluctuations. Under CS, phase transitions in membrane lipids increase membrane permeability and reduce fluidity, which are major causes of cold-induced damage in plants [31]. To counter this, plants have developed adaptive mechanisms such as membrane lipid remodeling and enhanced fatty acid desaturation [46,47,48,49,50,51,52]. Carbohydrates are crucial to potato responses under low-temperature stress. Expression of sucrose:sucrose 1-fructosyltransferase gene enhances freezing tolerance and increases soluble carbohydrate levels in transgenic plants compared to those of wild-type (WT) and antisense plants [53]. The differences between our findings and those of previous studies may result from conservation and evolutionary selection. Potatoes may remodel various metabolites in response to CS. The role of these metabolites in cold tolerance can be further verified through exogenous application or genetic manipulation of their biosynthetic pathways, including gene overexpression or knockout.

### 4.3. Investigation of Transcription Factors in the Cold Stress Response of Potatoes

TFs are key molecular regulators of plant cold resistance. In this study, MYB, bHLH, and WRKY TFs were identified as major regulators in the potato response to low-temperature stress. TFs regulate metabolite synthesis by regulating the expression of different structural genes, consistent with our findings. Comparative analysis of *ZmMYB31*-overexpressing from *Arabidopsis* and WT lines reveals that the maize (*Zea mays* L.) gene *ZmMYB31* enhances chilling and oxidative stress tolerance by modulating cold-responsive genes, thereby reducing ion leakage, ROS accumulation, and low-temperature photoinhibition [54]. In Rosa persica, RbebHLH may mediate low-temperature responses through the jasmonic acid signaling pathway [55]. Overexpressing *PmWRKY57* from *Prunus mume* in *Arabidopsis thaliana* enhances cold tolerance compared to WT plants. Under CS, transgenic *Arabidopsis thaliana* leaves exhibited higher SOD and POD activities and greater proline content than those of WP plants. Furthermore, these transgenic lines show significantly increased expression of cold-responsive genes (*AtCOR6.6*, *AtCOR47*, *AtKIN1*, and *AtRCI2A*) in their leaves [56]. To withstand harsh conditions, many plants employ TFs to regulate target gene expression and enhance CS tolerance [57]. Therefore, future studies should use genetic modification to further explore the functions of key candidate TFs.

### 4.4. Key Kyoto Encyclopedia of Genes and Genomes Pathways Identified Through the Combined Analysis

KEGG pathway analysis of DAMs and DEGs revealed significant enrichment in several metabolic pathways, particularly those related to flavonoids and core metabolic pathways involving lipids, amino acids, carbohydrates, nucleotides, and energy metabolism in KY130 vs. CS-KY130 and KY140 vs. CS-KY140. The coordinated activation of these metabolites and genes across various metabolic pathways highlights the cold defense mechanisms in potatoes. Lv et al. [28] integrated and analyzed DAMs and DEGs between Jimai325 and MU-134 after CS, revealing enrichment of the flavonol biosynthesis pathway (including “phenylpropanoid biosynthesis,” “flavonoid biosynthesis,” and “flavone and flavonol biosynthesis”), and amino acid biosynthesis pathways in wheat (*Triticum aestivum* L.). Following CS treatment in rapeseed (*Brassica napus* L.), functional enrichment analysis of DAMs and the corresponding DEGs revealed predominant enrichment in carbohydrate and amino acid metabolism [58]. Transcriptomic and metabolomic analyses of cold-sensitive (ZQ) and cold-tolerant (XL) *qingke* varieties under CS revealed significant enrichment of lipid metabolism pathways in response to freezing [52]. Therefore, future studies on these KEGG pathways are needed to clarify their regulatory mechanisms.

### 4.5. Flavonoid-Related Metabolism Plays an Important Role in Potato Response to Low Temperatures

DEGs encoding PAL, COMT, 4CL, CAD, and CYP75B1 were consistently upregulated in KY130 vs. CS-KY130 and KY140 vs. CS-KY140. In contrast, DEGs encoding flavonol synthase (FLS) were downregulated in KY130 vs. CS-KY130 but were upregulated and downregulated in KY140 vs. CS-KY140. During CS in potato, these enzyme activities fluctuate, potentially influencing downstream metabolites to enhance cold resistance. In peanut (*Arachis hypogaea* L.), genes, such as *PAL*, *4CL*, and *COMT*, showed higher transcript levels under CS in the cold-tolerant variety SLH than those in the cold-susceptible variety ZH12 [59]. Under CS, 59 of 83 genes involved in the phenylpropanoid/flavonoid metabolic pathways (Ko00940, Ko00941, and Ko00944) were significantly upregulated in wheat. These included genes encoding key enzymes (CYP73, 4CL, PAL, and CSE). Furthermore, CYP75B1 plays a critical role in the CS-related pathways Ko00941 and Ko00944 in wheat [28]. In response to low temperature, key enzyme genes, including COMT and CAD, were upregulated and more highly expressed in the “Donghe No. 1” (cold-tolerant) than in the “21st Century” (cold-sensitive), suggesting their major role in peach cold adaptation [41]. In tea (*Camellia sinensis* L.), low temperature induces *CsFLS* expression [60]. Among the regulatory mechanisms, enzymes play crucial roles in these complex processes. However, their specific functions in potatoes require further investigation.

Changes in butin levels mirrored those of *StCYP75B1*(*Soltu.Atl.08_1G014450*) in KY130 vs. CS-KY130 and KY140 vs. CS-KY140, suggesting a critical role for the flavonoid metabolic pathway in potato responses to low temperature. Promoters of *StCYP75B1*(*Soltu.Atl.08_1G014450*) and *StPAL*(*Soltu.Atl.06_2G003190*, *Soltu.Atl.09_3G003640*, and *Soltu.Atl.10_2G008250*) contained MYB cis-acting elements. Similarly, promoters of *StHCT*(*Soltu.Atl.01_1G026080*) contained HSF cis-acting elements, and *StCAD*(*Soltu.Atl.03_2G008390*) contained bHLH cis-acting elements, and *StHCT*(*Soltu.Atl.S021380*) contained a NAC cis-acting element [https://plantregmap.gao-lab.org/binding_site_prediction.php (accessed on 14 August 2025)]. The functions of these low-temperature-related TFs and structural genes require further validation via transgenic experiments.

### 4.6. Limitations

While this study shows the potential role of caffeic acid in potato cold resistance through combined metabolomic and transcriptomic analyses, and clarifies the molecular mechanism of CS response, some limitations remain.

First, the coverage of research samples and dynamic processes was insufficient. The types of materials and sample sizes were relatively small, and analyses were performed at only a single time point. This limits general conclusions on cold resistance mechanisms across potato germplasm and obscures dynamic changes in caffeic acid metabolism and its regulatory network during CS.

Second, the key findings require independent verification and functional confirmation. The reliability of the identified cold-resistant metabolites (e.g., caffeic acid) and genes must be verified in larger, independent sample sets. The specific biological functions of these candidate molecules in potato cold resistance remain unconfirmed through approaches such as exogenous application (e.g., spraying caffeic acid) or genetic transformation (e.g., gene overexpression/knockout).

Third, the analysis of upstream and downstream molecular mechanisms remains incomplete. The current findings reveal correlation patterns, but the molecular mechanisms—such as how low-temperature signals induce caffeic acid accumulation, its specific downstream targets, and the upstream regulatory network of key genes—remain unclear.

Lastly, limitations in the distinction between causality and association are present. Genomic analysis primarily reveals associations rather than causality. Although caffeic acid is strongly implicated in cold resistance, without functional experimental intervention, it remains possible that its accumulation is merely a correlated response rather than a causal factor.

Based on these limitations, future research should focus on two directions:

First, expand the breadth and depth study by validating findings across more potato varieties with varying cold resistance and multiple stress periods, to comprehensively assess cold resistance mechanisms.

Second, conduct functional verification experiments using approaches such as the exogenous application of substances (e.g., caffeic acid) or genetic transformation (e.g., gene overexpression/knockout). This would directly confirm the causal roles of these molecules in the formation of cold-resistant phenotypes and elucidate their mechanisms of action.

## 5. Conclusions

In this study, we conducted physiological, metabolomic, and transcriptomic analyses to identify key mechanisms of cold resistance in two potato varieties. Across all treatments, KY130 showed significantly higher antioxidant enzyme activities (CAT, SOD) and osmolyte levels (proline, soluble protein, soluble sugar) than those of KY140 (Figure 2C–Q). ELISA analysis revealed that caffeic acid levels were higher in KY130 than in KY140, and in CS-KY130 than in CS-KY140 (Figure 2A). These findings align with metabolomic trends. Following CS, KY130 and KY140 showed enrichment of lipids, flavonoids, terpenoids, phenolic acids, organic acids, alkaloids, lignans and coumarins, amino acids and derivatives, nucleotides and derivatives, and other metabolites (Figure 3B,E). KEGG pathway analysis of DAMs and DEGs revealed significant enrichment in flavonoid-related and core metabolic pathways, including lipids, amino acids, carbohydrates, nucleotides, and energy (Figure 8). Overall, these findings advance our understanding of caffeic acid and the molecular mechanisms underlying CS responses in potatoes.

## Figures and Tables

**Figure 1 plants-14-03447-f001:**
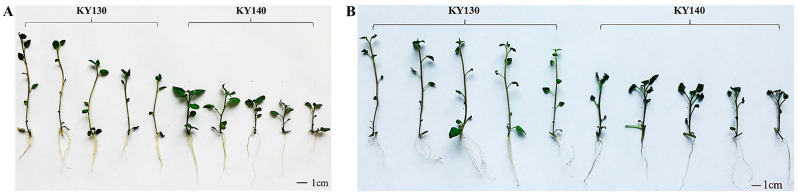
Differential physiological responses of potato cultivars KY130 and KY140 under CS. (**A**) Phenotypic characteristics of KY130 and KY140 under control conditions. (**B**) Phenotypic characteristics of CS-KY130 and CS-KY140. KY130, untreated KY130; CS-KY130, treated KY130; KY140, untreated KY140; CS-KY140, treated KY140. **Abbreviations:** CS, cold stress; KY130, potato cultivar KY130; KY140, potato cultivar KY140; CS-KY130, cold-stressed KY130; CS-KY140, cold-stressed KY140.

**Figure 2 plants-14-03447-f002:**
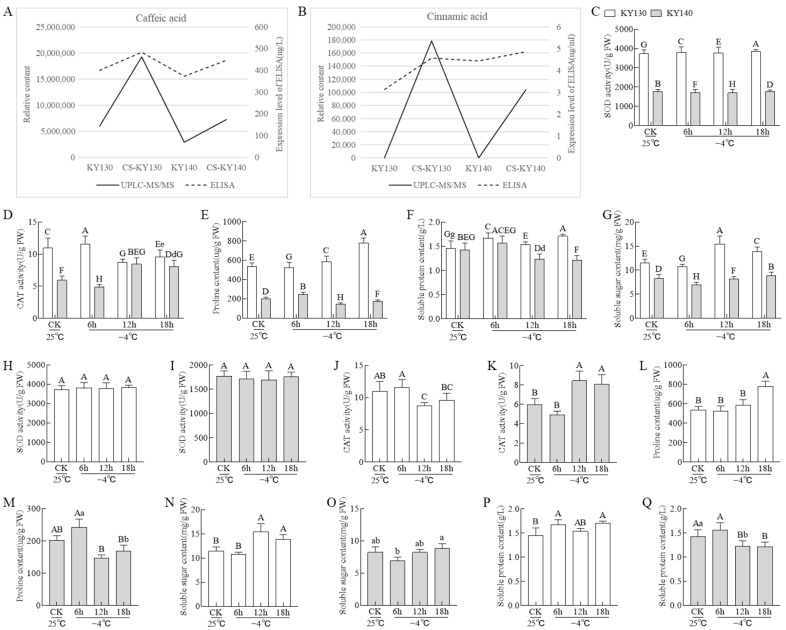
Biochemical and physiological responses of potato cultivars to CS. (**A**,**B**) Small-molecule indicators. KY130, untreated KY130; CS-KY130, treated KY130; KY140, untreated KY140; CS-KY140, treated KY140. (**C**–**G**) Comparisons of differences among groups. (**H**–**Q**) Comparisons of differences within groups. Bar heights represent the mean ± SD. Columns marked with different uppercase or lowercase letters indicate significant differences (*p* < 0.01 and *p* < 0.05, respectively). When uppercase and lowercase letters are present, lowercase letters denote significance at *p* < 0.05. **Abbreviations:** CS, cold stress; KY130, potato cultivar KY130; KY140, potato cultivar KY140; CS-KY130, cold-stressed KY130; CS-KY140, cold-stressed KY140; SD, standard deviation.

**Figure 3 plants-14-03447-f003:**
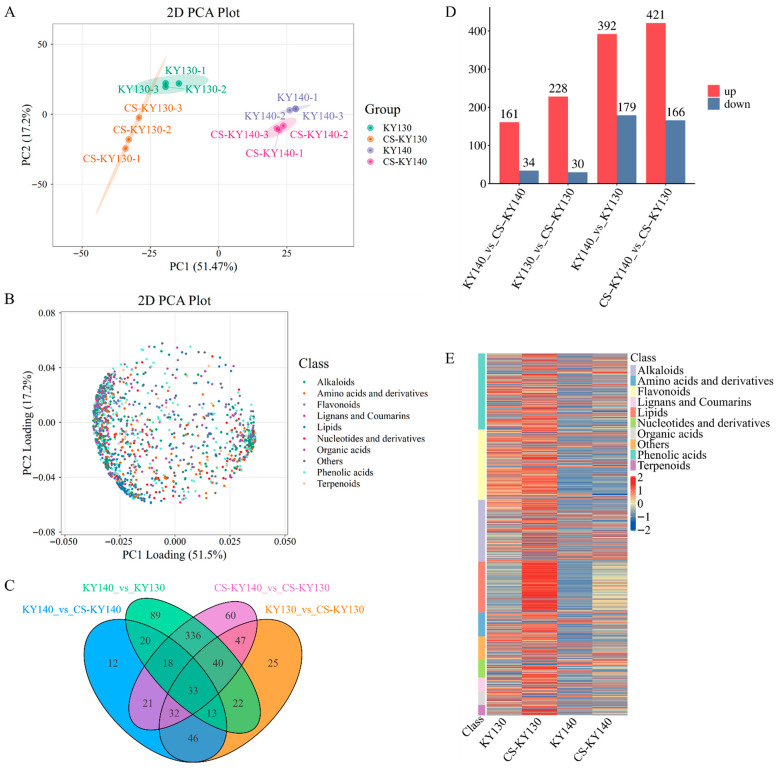
Metabolomic analysis of two potato cultivars exposed to CS. (**A**) PCA of metabolomic data derived from KY130 and KY140 in response to CS. (**B**) PCA loading plot of the same data, showing the contribution of different metabolite classes. (**C**) Venn diagram illustrating DAMs identified across four comparisons: KY140 vs. CS-KY140, KY130 vs. CS-KY130, KY140 vs. KY130, and CS-KY140 vs. CS-KY130. (**D**) Number of DAMs detected across the four comparisons (KY140 vs. CS-KY140, KY130 vs. CS-KY130, KY140 vs. KY130, and CS-KY140 vs. CS-KY130). (**E**) Heatmap showing the union of all DAMs detected across the four comparisons. “Class” indicates the first-level classification of DAMs. The horizontal axis represents cultivar names, and the vertical axis lists DAMs. Different colors represent relative metabolite abundance after standardization (red, higher content; blue, lower content). KY130, untreated KY130; CS-KY130, treated KY130; KY140, untreated KY140; CS-KY140, treated KY140. **Abbreviations:** CS, cold stress; KY130, potato cultivar KY130; KY140, potato cultivar KY140; CS-KY130, cold-stressed KY130; CS-KY140, cold-stressed KY140; PCA, principal component analysis; DAMs, differentially accumulated metabolites.

**Figure 4 plants-14-03447-f004:**
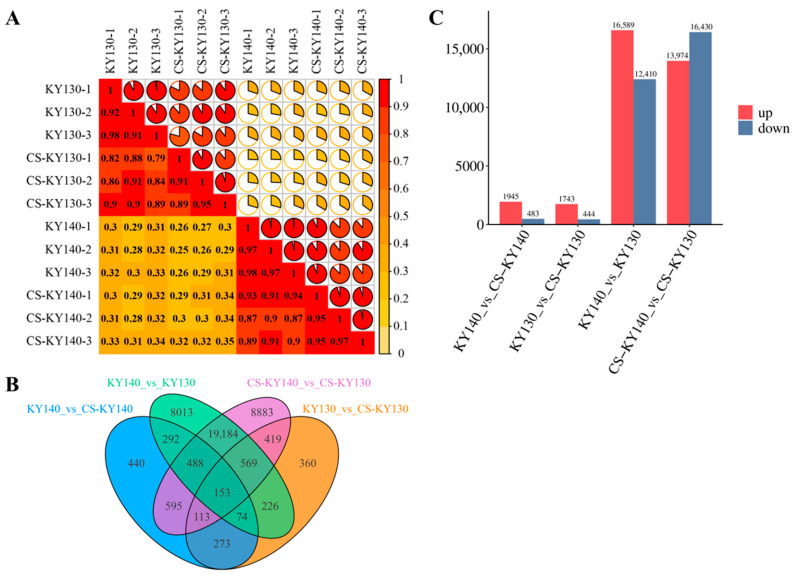
Transcriptomic analysis of two potato cultivars exposed to CS. (**A**) Correlation analysis of transcriptomic data across all experimental samples. (**B**) Venn diagram showing DEGs identified across four comparisons: KY140 vs. CS-KY140, KY130 vs. CS-KY130, KY140 vs. KY130, and CS-KY140 vs. CS-KY130. (**C**) Number of DEGs detected across the four comparisons (KY140 vs. CS-KY140, KY130 vs. CS-KY130, KY140 vs. KY130, and CS-KY140 vs. CS-KY130). KY130, untreated KY130; CS-KY130, treated KY130; KY140, untreated KY140; CS-KY140, treated KY140. **Abbreviations:** CS, cold stress; KY130, potato cultivar KY130; KY140, potato cultivar KY140; CS-KY130, cold-stressed KY130; CS-KY140, cold-stressed KY140; DEGs, differentially expressed genes.

**Figure 5 plants-14-03447-f005:**
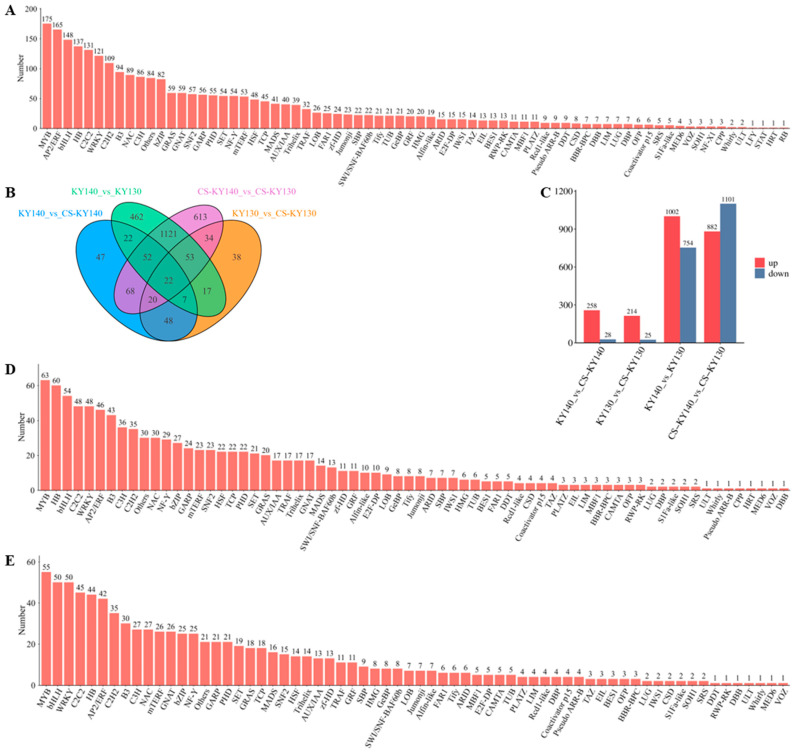
Analysis of TFs across four comparisons: KY140 vs. CS-KY140, KY130 vs. CS-KY130, KY140 vs. KY130, and CS-KY140 vs. CS-KY130. (**A**) Classification of TF families (bar chart) across the four comparisons (KY140 vs. CS-KY140, KY130 vs. CS-KY130, KY140 vs. KY130, and CS-KY140 vs. CS-KY130). (**B**) Venn diagram showing differentially expressed TFs identified across the four comparisons (KY140 vs. CS-KY140, KY130 vs. CS-KY130, KY140 vs. KY130, and CS-KY140 vs. CS-KY130). (**C**) Number of differentially expressed TFs detected across the four comparisons. KY130, untreated KY130; CS-KY130, treated KY130; KY140, untreated KY140; CS-KY140, treated KY140. (**D**) Analysis of TFs upregulated in KY140 vs. KY130 and in (**E**) CS-KY140 vs. CS-KY130. **Abbreviations:** TFs, transcription factors; KY130, potato cultivar KY130; KY140, potato cultivar KY140; CS-KY130, cold-stressed KY130; CS-KY140, cold-stressed KY140.

**Figure 6 plants-14-03447-f006:**
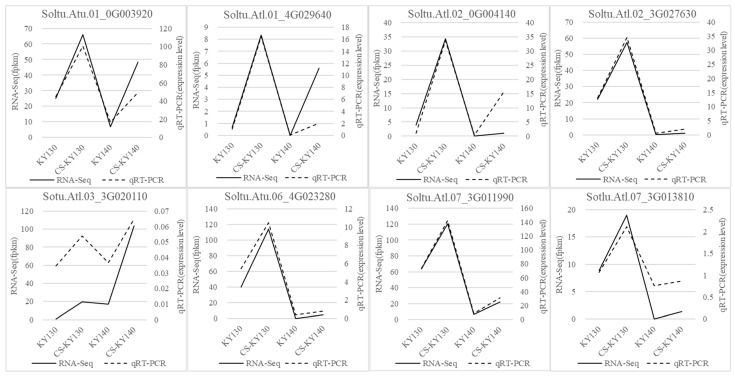
CS-induced expression patterns of eight DEGs in potato cultivars. KY130, untreated KY130; CS-KY130, treated KY130; KY140, untreated KY140; CS-KY140, treated KY140. **Abbreviations:** CS, cold stress; KY130, potato cultivar KY130; KY140, potato cultivar KY140; CS-KY130, cold-stressed KY130; CS-KY140, cold-stressed KY140; DEGs, differentially expressed genes.

**Figure 7 plants-14-03447-f007:**
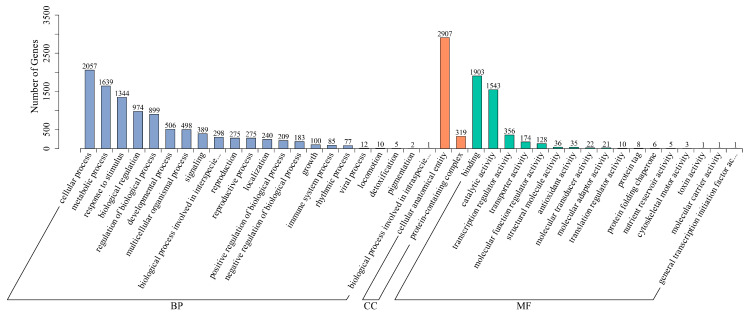
GO enrichment analysis of DEGs under CS in the comparisons KY130 vs. CS-KY130 and KY140 vs. CS-KY140. KY130, untreated KY130; CS-KY130, treated KY130; KY140, untreated KY140; CS-KY140, treated KY140. **Abbreviations:** GO, Gene Ontology; CS, cold stress; KY130, potato cultivar KY130; KY140, potato cultivar KY140; CS-KY130, cold-stressed KY130; CS-KY140, cold-stressed KY140; DEGs, differentially expressed genes.

**Figure 8 plants-14-03447-f008:**
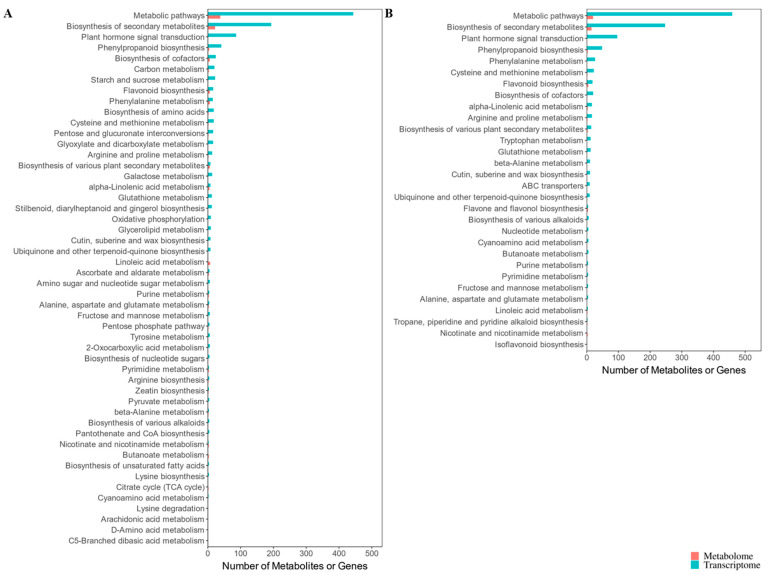
Integrated KEGG pathway analysis of DAMs and DEGs. (**A**) KY130 vs. CS-KY130 DAMs and DEGs common KEGG enrichment bar. (**B**) KY140 vs. CS-KY140 DAMs and DEGs common KEGG enrichment bar. The overlapping KEGG pathways identified between the two omics datasets are presented as bar charts displaying the number of DAMs and DEGs enriched within each pathway. When > 50 KEGG pathways were shared, only the top 50—ranked according to *p*-value with reference to the transcriptomic data—are shown. When < 50 enriched pathways are identified, all are displayed. The length of each bar (x-axis) indicates DAM/DEG counts, while their position (y-axis) shows the corresponding KEGG pathway. KY130, untreated KY130; CS-KY130, treated KY130; KY140, untreated KY140; CS-KY140, treated KY140. **Abbreviations:** KEGG, Kyoto Encyclopedia of Genes and Genomes; KY130, potato cultivar KY130; KY140, potato cultivar KY140; CS-KY130, cold-stressed KY130; CS-KY140, cold-stressed KY140; DAMs, differentially accumulated metabolites; DEGs, differentially expressed genes; CS, cold stress.

**Figure 9 plants-14-03447-f009:**
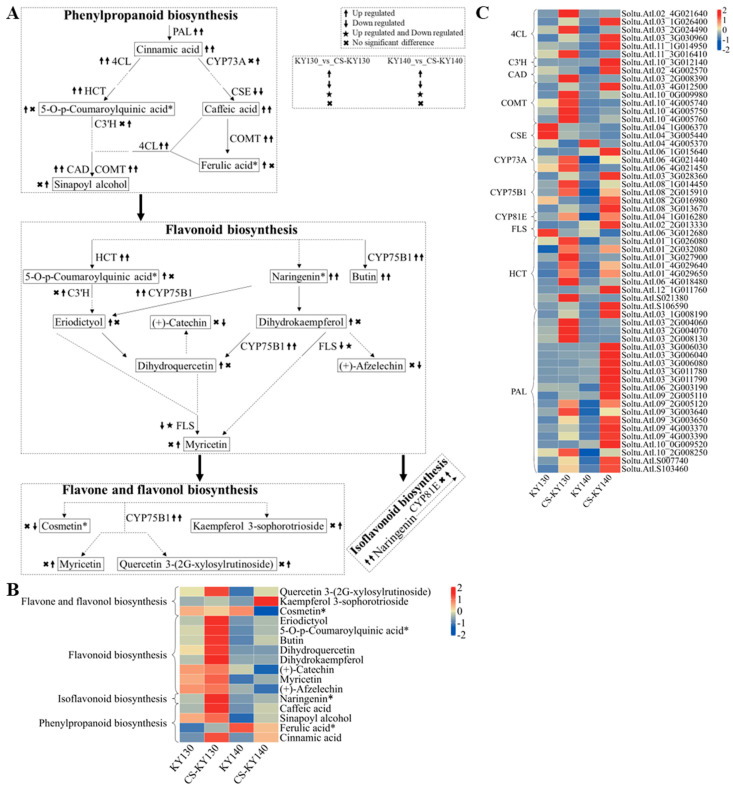
Integrated metabolomic and transcriptomic analysis of DAMs and DEGs associated with “flavonoid-related metabolism in potatoes under CS. (**A**) Schematic representation of the “flavonoid-related metabolic pathway.” Arrow-headed lines connect the DAMs at both ends, with enzymes positioned along each line. Solid lines indicate direct interactions between DAMs, while dashed lines represent indirect connections through intermediate metabolites. DAMs and DEGs were analyzed through KEGG pathway mapping to elucidate their potential roles in systemic biological functions [https://www.kegg.jp/kegg/pathway.html (accessed on 8 July 2024)]. (**B**) Heatmap of DAMs associated with flavonoid-related metabolism. The horizontal axis represents the sample names, while the vertical axis denotes the KEGG pathway. To prevent redundancy, DAMs involved in multiple pathways are presented within a single representative pathway. Asterisks (*) indicate the presence of metabolite isomers. Distinct colors denote relative metabolite abundance after standardization (red, higher content; blue, lower content). (**C**) Heatmap illustrating DEGs encoding key enzymes involved in flavonoid-related metabolism. The horizontal axis represents the sample names, while the vertical axis denotes the corresponding enzymes and DEGs. Distinct colors denote standardized relative expression levels (red, higher content; blue, lower content). KY130, untreated KY130; CS-KY130, treated KY130; KY140, untreated KY140; CS-KY140, treated KY140. **Abbreviations:** KEGG, Kyoto Encyclopedia of Genes and Genomes; KY130, potato cultivar KY130; KY140, potato cultivar KY140; CS-KY130, cold-stressed KY130; CS-KY140, cold-stressed KY140; DAMs, differentially accumulated metabolites; DEGs, differentially expressed genes; CS, cold stress; PAL, phenylalanine ammonia-lyase; 4CL, 4-coumarate:CoA ligase; HCT, shikimate O-hydroxycinnamoyltransferase; CYP73A, trans-cinnamate 4-monooxygenase; CSE, caffeoyl shikimate esterase; C3′H, 5-O-(4-coumaroyl)-D-quinate 3′-monooxygenase; CAD, cinnamyl-alcohol dehydrogenase; COMT, caffeic acid 3-O-methyltransferase; CYP75B1, flavonoid 3′-monooxygenase; FLS, flavonol synthase; CYP81E, isoflavone/4′-methoxyisoflavone 2′-hydroxylase.

**Figure 10 plants-14-03447-f010:**
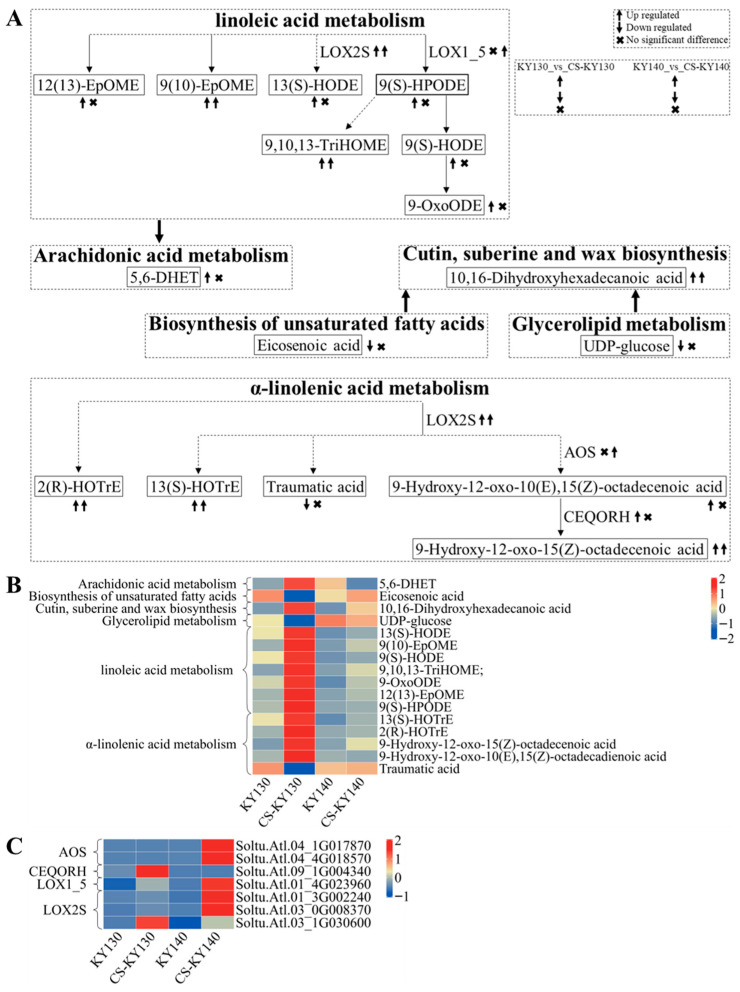
Integrated metabolomic and transcriptomic analysis of DAMs and DEGs associated with “lipid metabolism” in potatoes under CS. (**A**) Schematic representation of the “lipid metabolic pathway.” Arrow-headed lines indicate connections between DAMs, with enzymes positioned along each line. Solid lines indicate direct interactions between DAM molecules, while dashed lines represent indirect connections with intervening metabolites. DAMs and DEGs were analyzed through KEGG pathway mapping to elucidate their potential roles in systemic biological processes [https://www.kegg.jp/kegg/pathway.html (accessed on 8 July 2024)]. (**B**) Heatmap of DAMs associated with lipid metabolism. The horizontal axis represents sample names, while the vertical axis corresponds to the KEGG pathway. Colors represent different values obtained following the standardization of relative contents (red, higher content; blue, lower content). (**C**) Heatmap depicting DEGs encoding key enzymes involved in “lipid metabolism.” The horizontal axis represents sample names, while the vertical axis denotes the corresponding enzymes and DEGs. Distinct colors represent standardized values of various relative contents (red, higher content; blue, lower content). KY130, untreated KY130; CS-KY130, treated KY130; KY140, untreated KY140; CS-KY140, treated KY140. **Abbreviations:** KEGG, Kyoto Encyclopedia of Genes and Genomes; KY130, potato cultivar KY130; KY140, potato cultivar KY140; CS-KY130, cold-stressed KY130; CS-KY140, cold-stressed KY140; DAMs, differentially accumulated metabolites; DEGs, differentially expressed genes; CS, cold stress; LOX2S, lipoxygenase; LOX1_5, linoleate 9S-lipoxygenase; AOS, hydroperoxide dehydratase; CEQORH, chloroplastic oxoene reductase.

**Figure 11 plants-14-03447-f011:**
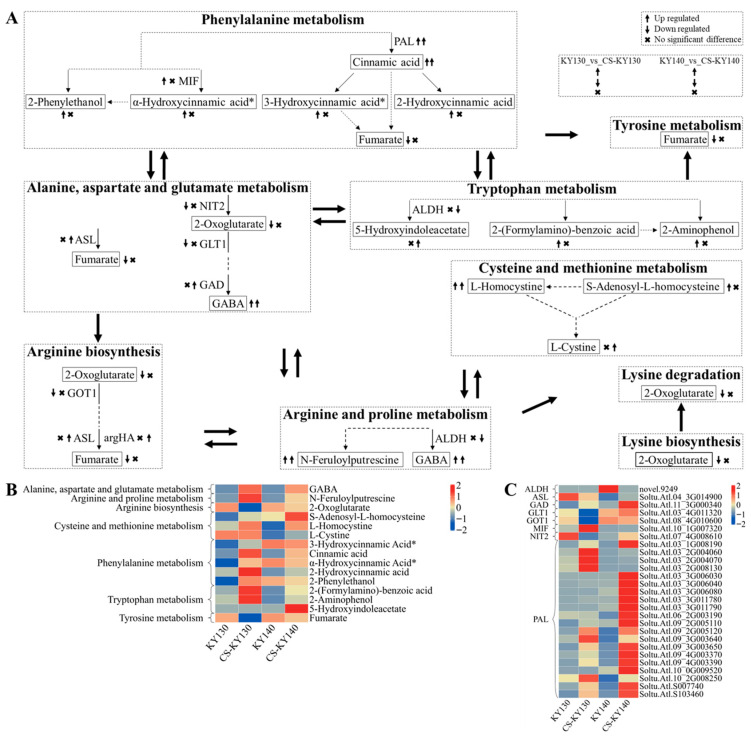
Integrated metabolomic and transcriptomic analysis of DAMs and DEGs associated with “amino acid metabolism” in potatoes under CS. (**A**) Schematic representation of the “amino acid metabolic pathway.” Arrow-headed lines connect DAMs, with enzymes positioned along each line. Solid lines indicate direct interactions between DAM molecules, while dashed lines represent indirect connections with intervening metabolites. DAMs and DEGs were analyzed through KEGG pathway mapping to elucidate their potential roles in systemic biological functions [https://www.kegg.jp/kegg/pathway.html (accessed on 8 July 2024)]. (**B**) Heatmap of DAMs associated with amino acid metabolism. The horizontal axis represents sample names, while the vertical axis corresponds to the KEGG pathway. To prevent redundancy, DAMs involved in multiple pathways are shown in a single representative pathway. Asterisks (*) represent the presence of metabolite isomers. Colors indicate standardized values of the relative contents (red, higher content; blue, lower content). (**C**) Heatmap illustrating DEGs encoding key enzymes involved in amino acid metabolism. The horizontal axis represents sample names, while the vertical axis corresponds to the enzymes and DEGs. Distinct colors represent the standardized values of various relative contents (red, higher content; blue, lower content). KY130, untreated KY130; CS-KY130, treated KY130; KY140, untreated KY140; CS-KY140, treated KY140. **Abbreviations:** KEGG, Kyoto Encyclopedia of Genes and Genomes; KY130, potato cultivar KY130; KY140, potato cultivar KY140; CS-KY130, cold-stressed KY130; CS-KY140, cold-stressed KY140; DAMs, differentially accumulated metabolites; DEGs, differentially expressed genes; CS, cold stress; MIF, Phenylpyruvate tautomerase; PAL, phenylalanine ammonia-lyase; ASL, argininosuccinate lyase; NIT2, omega-amidase; GLT1, glutamate synthase (NADH); GAD, glutamate decarboxylase; GOT1, aspartate aminotransferase, cytoplasmic; argHA, argininosuccinate lyase/amino-acid N-acetyltransferase; ALDH, aldehyde dehydrogenase (NAD+).

**Table 1 plants-14-03447-t001:** Primer sequence.

Gene ID	Forward Primer (5′-3′)	Reverse Primer (5′-3′)
*Soltu.Atl.01_0G003920*	CACAGGCACAACTTGCACTC	GACGCCATCGAACCTGAGAA
*Soltu.Atl.01_4G029640*	CGAATCATCTTCAGAAGCTTTCAT	ACTGCAGCTTGGCAAGTAGT
*Soltu.Atl.02_0G004140*	TCCCCTGCTAATTCATGGCG	TCACACACTTGGGATTCAGCA
*Soltu.Atl.02_3G027630*	CCTTGGCTGCTGATTCCAGA	AAGCAGTGTGATGCCTTCCA
*Soltu.Atl.03_3G020110*	GTGCTGGAGTTGCAGTACCT	AGCTGAGTCTTCGACGTACC
*Soltu.Atl.06_4G023280*	AGGTGCGAGTTAATGGTCCG	TGCTGGAACCGAGAAAACCA
*Soltu.Atl.07_3G011990*	AGAGGCAGAGGAAAGTACACA	AAGCACTTGCAATTGGATCAC
*Soltu.Atl.07_3G013810*	CAAATTCACTCGCACACGGC	AGCGTGCGTCAACAGTAAAG
*StActin*	AGATGCTTACGCTGGATGGAATGC	TTCCGGTGTGGTTGGATTCTGTTC

**Table 2 plants-14-03447-t002:** Reaction procedure of qRT-PCR.

Preincubation (1 Cycle)	2 Step Amplification (45 Cycle)	Melting (1 Cycle)	Cooling (1 Cycle)
95 °C for 600 s	95 °C for 10 s	95 °C for 10 s	37 °C for 30 s
	Tm°C for 30 s	65 °C for 60 s	
		97 °C for 1 s	

Tm, primer annealing temperature; qRT-PCR, quantitative real-time polymerase chain reaction.

## Data Availability

The datasets employed in this study are publicly available via online repositories. The repository/repositories and associated accession number(s) are available at https://www.ncbi.nlm.nih.gov/geo/query/acc.cgi?acc=GSE291340 (accessed on 6 March 2025).

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
