# Peer review of "Integrated Metabolomics and Transcriptomics Analyses Reveal the Critical Role of Caffeic Acid in Potato (*Solanum tuberosum* L.) Cold Tolerance"

_plants, 2025, doi:10.3390/plants14223447_

Round 1

Reviewer 1 Report

Comments and Suggestions for Authors

Comments

The abstract section requires revision for greater clarity and conciseness, particularly in terms of grammar and redundant phrasing.

Grammar and Phrasing (Lines 15-16): Change the phrase "ELISA analysis revealed that caffeic acid levels in KY130 exhibited higher than KY140" to "ELISA analysis revealed higher caffeic acid levels in KY130 compared to KY140."

Line 19-21: Change the phrase "… might both related to direct cold resistance of potato and involved in regulating potato cold resistance" to "… were found to be directly related to cold resistance and involved in its regulation"

Introduction:

The introduction section is very long, disorganized, and contains unclear sentences. In general, it needs a substantial rewriting, but some of the points that need improvement are listed below:

  1. Begin the introduction section by highlighting the general importance of the potato crop and the challenge of cold stress in its production. First, introduce the core problem of how cold stress limits potato growth and development.
  2. Avoid a long list of examples; instead, concisely introduce the transcriptomics and metabolomics approaches as powerful tools for understanding complex plant stress responses in general and cold stress in particular.
  3. Following the establishment of the general methodology, focus specifically on potatoes and explain that the specific mechanisms of cold tolerance in potato varieties are not fully understood.
  4. Please try to connect the Previous work with your current study, NOT repeating their findings, and summarize their result
  5. Please conclude the introduction with a clear statement of the study's purpose and its expected contribution

Methods section

The methods section is difficult to understand and contains several inconsistencies and grammatical errors. Some of the points are listed below.

  1. It contains conflicting information on the cold treatment duration (eg., lines 134-135 state the cold treatment was for 7 days. Line 143 states it was for 12 hours. Line 145 states that for physiological parameters, it was 6, 12, and 18 hours. Lines 147-148 state that for the multi-omics analysis, it was 12 hours. There is a mix-up of cold treatment from 7 days to different hours for the different parameters. This is difficult to follow, and it should specify clearly which experiment corresponds to each duration.
  2. The recovery period from the cold treatment is given as a range (12-24h), which is not precise. It needs further explanation why this much range is needed.
  3. Several sentences (eg., line 136-140) are grammatically incorrect and need revision.
  4. The specific type of MS medium composition is not clearly mentioned (e.g., half-strength, full-strength).
  5. What are the measured physiological parameters, and which specific instrument is used?
  6. In line 157, the title is "OMICS analysis". In general, the omics analysis refers to the comprehensive study of genes, proteins, and metabolites to understand their interactions and how they contribute to biological systems. But you did only RNAseq analysis and some metabolite classes. Omics analysis is a broad term but the scale of your analysis is not as broad as you mentioned in the title. Please revise this. In addition, the detailed methods for metabolomic and RNAseq are not described clearly rather it is mixed. Please describe the platform you used, sample preparation, .....
  7. For qPCR analysis, the details (like., primers, genes, reference genes, thermocycling conditions etc…) are missing.

Result:

The writing style in the results section is weak and problematic to follow, and needs a substantial re-writing. Below is a list of points that need to be addressed

  1. Line 187-189: "Potato thrives best …… Cold stress is one of the important environmental factors affecting the growth and development of potato species" is an introductory statement that belongs in the introduction section, not the results section. This section should focus only on the findings of your study.
  2. The phrase "7 days of 7-day cold stress (CS)" is repetitive and grammatically incorrect. Please revise.
  3. Figure 2 title: It is not the right way of presenting a figure title as "second line of pictures and the first picture from the left of the third line….". Please label A, B, C… for the different sections of the figure.
  4. Section 3.3: The PCA plot figure does not clearly presented how the samples are clustered, which metabolite classes are responsible for the separation. Start with the Big Picture by clearly referencing the PCA and correlation plots to show data quality.
  5. What are the Differentially Accumulated Metabolites (DAMs), were the change statistically significant, it lacks clarity.
  6. Please explain the difference between "direct cold resistance" and "regulating cold resistance" mean.
  7. Instead of listing all the identified metabolites, it is better to highlight only the most significant ones, perhaps those with the largest fold-changes, and discuss their potential roles in cold resistance based on existing literature.
  8. The provided numbers are raw counts (e.g., 195 DAMs) which lacking Quantitative Context and unexplained. The mentioned relative content values (e.g., 10009611, 2601061) are meaningless without a statistical analysis to show their significance or a fold-change for comparison to other metabolites.
  9. The last paragraph of section 3.3. begin with correlation analysis and describing about figure 4A but the later sentences present about the number of Differentially Expressed Genes (DEGs) without any organization or visual aid. In the first place nothing is written about how the genes are correlating each other and it simply lists counts of up/down-regulated genes in a single, dense block of text. This is not good way to present complex transcriptomic data. It needs major revision.
  10. The author states that "A total of 40082 DEGs were identified in the comparisons of all treatment combinations" but the total sum of the DEGs in all comparisons is much higher (64018) than the mentioned amount. This indicates an error in how the total was calculated.
  11. Section 3.5: The text is a list of raw numbers for different transcription factor (TF) families and the counts of up- and down-regulated TFs in various comparisons. This section is difficult to digest and it simply lists the number of TFs without discussing which ones are most significant in relation to cold tolerance.
  12. Section 3.9: The section contains long list of metabolites and genes without logical flow or statistical analysis. The away metabolites and genes presented is confusing and making it impossible to follow the implication.
  13. Some sentence in section 3.9 are confusing: eg., " Sinapoyl alcohol, Butin, Kaempferol 3-sophorotrioside were not significantly different in KY130_vs_CS-KY130, not detected in KY140, detected in CS-KY140."
  14. In this section (3.9), the authors present long list of genes and speculated their role in cold resistance rather than reporting a solid finding.
  15. Section 3.11: The section is difficult to follow because it contains a list of metabolite names and speculative statement without any clear flow. For example in the following sentence: " In this study, “amino acid metabolism” mainly include “arginine and proline metabolism”, “Phenylalanine metabolism”, “tryptophan metabolism”. L-Cystine might both related to direct cold resistance of potato and involved in regulating potato cold resistance". Long list of metabolism is mentioned and speculated by stating "might both related to direct cold resistance of potato and involved in regulating potato cold resistance ". The word "both" refers to two items but here you mentioned long lists and try to close up with both, it is not logical.

Discussion:

In most of the cases the discussion section is a repetition of the results section rather than interpret and explain the results. This section mainly relies on external citations to explain their findings but the discussion section should primarily interpret the result using citations to support their interpretations. The authors use inconsistent usage of both text and number citation format (eg., Line 506 and 610). The majority of this section is not properly organized and contains somehow disjointed facts rather than a cohesive scientific argument. In general the discussion section needs a thorough re-writing by focusing on explaining the biological significance of the findings rather than simply listing data and external citations.

Comments on the Quality of English Language

The text is poorly written; it requires professional English language editing. 

Author Response

We are deeply grateful to the reviewer for taking the time to provide such a thorough and constructive review of our manuscript. The comments are immensely helpful and have provided us with critical guidance for strengthening our paper. We have carefully addressed each comment and suggestion in the revised manuscript. Our detailed responses are listed below.

Comments

The abstract section requires revision for greater clarity and conciseness, particularly in terms of grammar and redundant phrasing.

Grammar and Phrasing (Lines 15-16): Change the phrase "ELISA analysis revealed that caffeic acid levels in KY130 exhibited higher than KY140" to "ELISA analysis revealed higher caffeic acid levels in KY130 compared to KY140."

Reply: corrected    Lines 15-16

Line 19-21: Change the phrase "… might both related to direct cold resistance of potato and involved in regulating potato cold resistance" to "… were found to be directly related to cold resistance and involved in its regulation"

Reply: corrected    Line 19-21

Introduction:

The introduction section is very long, disorganized, and contains unclear sentences. In general, it needs a substantial rewriting, but some of the points that need improvement are listed below:

  1. Begin the introduction section by highlighting the general importance of the potato crop and the challenge of cold stress in its production. First, introduce the core problem of how cold stress limits potato growth and development.

Reply: corrected   (The first paragraph of the introduction) Line 41-49

  1. Avoid a long list of examples; instead, concisely introduce the transcriptomics and metabolomics approaches as powerful tools for understanding complex plant stress responses in general and cold stress in particular.

Reply: corrected   (The third paragraph of the introduction) Line 61-72

  1. Following the establishment of the general methodology, focus specifically on potatoes and explain that the specific mechanisms of cold tolerance in potato varieties are not fully understood.

Reply: corrected    (The third paragraph of the introduction) Line 72-78

  1. Please try to connect the Previous work with your current study, NOT repeating their findings, and summarize their result

Reply: corrected    (the last paragraph) Line 79-80

  1. Please conclude the introduction with a clear statement of the study's purpose and its expected contribution

Reply: corrected  (the last paragraph) Line 87-92

Methods section

The methods section is difficult to understand and contains several inconsistencies and grammatical errors. Some of the points are listed below.

  1. It contains conflicting information on the cold treatment duration (eg., lines 134-135 state the cold treatment was for 7 days. Line 143 states it was for 12 hours. Line 145 states that for physiological parameters, it was 6, 12, and 18 hours. Lines 147-148 state that for the multi-omics analysis, it was 12 hours. There is a mix-up of cold treatment from 7 days to different hours for the different parameters. This is difficult to follow, and it should specify clearly which experiment corresponds to each duration.

Reply: 2.1.1 Experimental design for phenotyping with statistical evaluation    Line 100

-4℃ for 7 days

2.1.2 Measurement of caffeic acid and cinnamic acid levels using ELISA   Line 109

-4℃ for 12 h

2.1.3 Plant treatment and sampling for physiological assays   Line 113

-4°C for 6, 12, and 18 h

2.1.4 Sample preparation for metabolomic and transcriptomic analysis   Line 117

-4℃ for 12 h

  1. The recovery period from the cold treatment is given as a range (12-24h), which is not precise. It needs further explanation why this much range is needed.

Reply: Potato tissue culture seedlings grow faster in suitable environments. Potatoes were subjected to 7 days of cold stress (-4℃) and recovered at 22-25℃ before phenotypic identification was carried out. Our laboratory has identified more than 30 varieties using this method. Some varieties can identify phenotypes after 12 hours of recovery, while some varieties have poor recovery effect after 12 hours. If the recovery time is long, new branches and leaves will grow out of the potato tissue culture seedlings, which will affect the accuracy of the results. Recovery results at 12h and 24h are almost the same.

  1. Several sentences (eg., line 136-140) are grammatically incorrect and need revision.

Reply: Thank you for this suggestion. We highly value the importance of professional language polishing. In our current revision, we have focused primarily on addressing the substantial scientific points raised by all reviewers. Our plan is to perform a comprehensive language editing and proofreading of the entire manuscript once all the scientific revisions are finalized to ensure consistency and high quality. We will certainly do this before the final submission.

  1. The specific type of MS medium composition is not clearly mentioned (e.g., half-strength, full-strength).

Reply: MS medium (Carrageenan) (Shijiazhuang Nutrition Tissue Culture Technology Co., Ltd., Shijiazhuang, China) Line 96-97

  1. What are the measured physiological parameters, and which specific instrument is used?

Reply: Enzymatic activities (CAT, SOD) and osmolytes content (proline, soluble protein, sugar) were quantified using respective kits (Nanjing Jiancheng Biology Co., Ltd., Nanjing, China) on Ultraviolet-visible spectrophotometer (Unico (Shanghai) Instrument Co., Ltd., Shanghai, China). Line 126-129

  1. In line 157, the title is "OMICS analysis". In general, the omics analysis refers to the comprehensive study of genes, proteins, and metabolites to understand their interactions and how they contribute to biological systems. But you did only RNAseq analysis and some metabolite classes. Omics analysis is a broad term but the scale of your analysis is not as broad as you mentioned in the title. Please revise this. In addition, the detailed methods for metabolomic and RNAseq are not described clearly rather it is mixed. Please describe the platform you used, sample preparation, .....

Reply: corrected Line 130

  1. For qPCR analysis, the details (like., primers, genes, reference genes, thermocycling conditions etc…) are missing.

Reply: Table 1 Primer sequence  Line 169              Table 2  Reaction procedure of qRT-PCR  Line 170

Result:

The writing style in the results section is weak and problematic to follow, and needs a substantial re-writing. Below is a list of points that need to be addressed

  1. Line 187-189: "Potato thrives best …… Cold stress is one of the important environmental factors affecting the growth and development of potato species" is an introductory statement that belongs in the introduction section, not the results section. This section should focus only on the findings of your study.

Reply: deleted   Line 179

  1. The phrase "7 days of 7-day cold stress (CS)" is repetitive and grammatically incorrect. Please revise.

Reply: corrected    Line 181

  1. Figure 2 title: It is not the right way of presenting a figure title as "second line of pictures and the first picture from the left of the third line….". Please label A, B, C… for the different sections of the figure.

Reply: corrected   Figure 2  Line 198

  1. Section 3.3: The PCA plot figure does not clearly presented how the samples are clustered, which metabolite classes are responsible for the separation. Start with the Big Picture by clearly referencing the PCA and correlation plots to show data quality.

Reply: corrected   3.3  Line  205-209

  1. What are the Differentially Accumulated Metabolites (DAMs), were the change statistically significant, it lacks clarity.

Reply: The screening criteria for DAMs were established as follows: Variable importance in projection (VIP) ≥ 1 and fold change (FC) ≥ 2 or FC ≤ 0.5. DAMs refer to metabolites whose content changes significantly in different comparison groups. Line 141-143

  1. Please explain the difference between "direct cold resistance" and "regulating cold resistance" mean.

Reply: Change "regulating cold resistance" to “were directly related to the indirect cold resistance of potato”

The picture above is a master's thesis, English abstract(Zhang, Y. L. 2023).

The above content is the origin of this study “were directly related to the direct cold resistance of potato”。“were directly related to the direct cold resistance of potato” is the characteristics of the material itself, which is only related to the genetic background of the material itself, and is just a comparison between varieties that have not undergone any treatment.

Defining this metabolite as "were directly related to the direct cold resistance of potato" just if the content of cold-resistant materials is higher than the content of sensitive materials. The range of valuable metabolites given by this definition is too wide. So some restrictions were put in place:The screening criteria for DAMs were established as follows: Variable importance in projection (VIP) ≥ 1 and fold change (FC) ≥ 2 or FC ≤ 0.5. DAMs refer to metabolites whose content changes significantly in different comparison groups. Screening criteria for DAMs that might “were directly related to the direct cold resistance of potato”: Upregulated DAMs in KY140_vs_KY130. Line 141-146

Screening criteria for DEGs that might “were directly related to the direct cold resistance of potato”: similarly

There are two commonly used analysis methods for analyzing potato physiological indicators (SOD, CAT, POD, proline, soluble protein, soluble sugar)。One is content comparison. After stress, the physiological index content of cold-resistant materials was higher than that of sensitive materials (Xu et al., 2016; Wang M. X. et al., 2021; Duan et al., 2022)。The other is the comparison of changes before and after coercion. After stress, the increase in cold-resistant materials was higher than that in sensitive materials (Yang and Guo 2016)。

“were directly related to the indirect cold resistance of potato” is a comparison between potato varieties after cold stress.

Screening criteria DAMs that might “were directly related to the indirect cold resistance of potato”: Upregulated DAMs in CS-KY140_vs_CS-KY130. Line 146-158

Screening criteria DEGs that might “were directly related to the indirect cold resistance of potato”: similarly

Zhang, Y. L. (2023). Identification of cold hardiness related phenotypes and analysis of characteristic metabolites in eight potato genotypes. Huazhong Agricultural University, Wuhan.

Xu, J.; Zheng, X.; Yan, H.F.; Tang, X.H.; Xiong, J.; Wei, M.Z.; Qin, W.Z.; Li, W.L. (2016). Physiological responses of different potato varieties to cold stress at seedling stage. Journal of Southern Agriculture. 47, 1837-1943. doi:10.3969/j:issn.2095-1191.2016.11.1837

Wang, M.X.; Mei, C.; Song, Q.N.; Wang, H.J.; Wu, S.Y.; Feng, R.Y. Evaluation on cold tolerance of six potato seedlings in tissue culture under low temperature stress. J. Shanxi Agric. Sci. 2021, 49, 1502-1506. doi:10.3969/j.issn.1002-2481.2021.12.20

Duan, J. X., Hou, L. X., Bao, H. H., Wang, X. M., Zheng, H. J., Zhu, G. T., et al. (2022). Effects of low temperature stress on physiological indexes related to cold resistance of potato seedlings. J. Yunnan Normal Univ. (Natural Sci. Edition). 42, 20–26. doi: 10.7699/j.ynnu.ns-2022-058

Yang HJ and Guo HC.(2016) Physiological Responds of Potato Seedlings to Low Temperature Stress and Comprehensive Evaluation on Their Cold Tolerance. Southwest China Journal of Agricultural Sciences. DOI:10.16213/j.cnki.scjas.2016.11.009

  1. Instead of listing all the identified metabolites, it is better to highlight only the most significant ones, perhaps those with the largest fold-changes, and discuss their potential roles in cold resistance based on existing literature.

Reply: 3.3 It is mainly divided into 5 parts:Principle component analysis (PCA) (Figures 3A,B); Venn diagram and statistical plots that objectively describe the 4 combinations (KY140_vs_CS-KY140, KY130_vs_CS-KY130, KY140_vs_KY130 and CS-KY140_vs_CS-KY130) (Figures 3C, D); Objectively described the heat map of the union of all DAMs detected in 4 combinations (KY140_vs_CS-KY140, KY130_vs_CS-KY130, KY140_vs_KY130 and CS-KY140_vs_CS-KY130) (Figures 3E); Based on the criteria for screening “metabolites might linked to potato cold resistance” in this study, All soluble sugars might linked to potato cold resistance are introduced。Soluble sugars (Line 229-234) here correspond to the discussion (Line 527-534), Although 3.12 (Line 460) is soluble sugar metabolism, the soluble sugar described here is not in 3.12 soluble sugar metabolism. Although the most important metabolite in this study is caffeic acid, caffeic acid is introduced in 3.9. (Line 345)

  1. The provided numbers are raw counts (e.g., 195 DAMs) which lacking Quantitative Context and unexplained.

Reply:195 is not raw counts. 195 are differentially accumulated metabolites (DAMs).  The text content corresponds clearly to Figure 3

The mentioned relative content values (e.g., 10009611, 2601061) are meaningless without a statistical analysis to show their significance or a fold-change for comparison to other metabolites.

Reply: deleted

  1. The last paragraph of section 3.3. begin with correlation analysis and describing about figure 4A but the later sentences present about the number of Differentially Expressed Genes (DEGs) without any organization or visual aid. In the first place nothing is written about how the genes are correlating each other and it simply lists counts of up/down-regulated genes in a single, dense block of text. This is not good way to present complex transcriptomic data. It needs major revision.

Reply: All words correspond to pictures one by one

  1. The author states that "A total of 40082 DEGs were identified in the comparisons of all treatment combinations" but the total sum of the DEGs in all comparisons is much higher (64018) than the mentioned amount. This indicates an error in how the total was calculated.

Reply: It is unreasonable to add the numbers at the top of all columns in the statistical chart because there may be the same DEGs between groups, and it is reasonable to add the numbers in the Venn diagram.

  1. Section 3.5: The text is a list of raw numbers for different transcription factor (TF) families and the counts of up- and down-regulated TFs in various comparisons. This section is difficult to digest and it simply lists the number of TFs without discussing which ones are most significant in relation to cold tolerance.

Reply: Added 

1002 TFs belong to 68 TFs families, including MYB (63), HB (60), bHLH (54), WRKY (48), C2C2 (48), AP2/ERF (46), and others might be directly related to the direct cold resistance of potato.    Line 289-291

882 TFs belong to 66 TFs families, including MYB (55), bHLH (50), WRKY (50), C2C2 (45), HB (44), AP2/ERF (42), and others might be directly related to the indirect cold resistance of potato.    Line 292-294

  1. Section 3.9: The section contains long list of metabolites and genes without logical flow or statistical analysis. The away metabolites and genes presented is confusing and making it impossible to follow the implication.

Reply: This paragraph mainly objectively describes metabolites or enzymes. Based on the criteria for screening “metabolites might linked to potato cold resistance” in this study to selecte metabolites. Line 144-148

Based on the criteria for screening “gene might linked to potato cold resistance” in this study to selecte gene. Line 154-158

  1. Some sentence in section 3.9 are confusing: eg., " Sinapoyl alcohol, Butin, Kaempferol 3-sophorotrioside were not significantly different in KY130_vs_CS-KY130, not detected in KY140, detected in CS-KY140."

Reply: The screening criteria for DAMs were established as follows: Variable importance in projection (VIP) ≥ 1 and fold change (FC) ≥ 2 or FC ≤ 0.5. DAMs refer to metabolites whose content changes significantly in different comparison groups. (Line 141-143).   Sinapoyl alcohol, Butin, Kaempferol 3-sophorotrioside do not meet these criteria in KY130_vs_CS-KY130. however, these metabolites meet these criteria in KY140_vs_CS-KY140. These metabolites not detected in KY140, detected in CS-KY140. This sentence is mainly an objective description of the facts, highlighting the special features of these metabolites: for example, these metabolites appeared after cold stress in KY140 and were synthesized at low temperature.

  1. In this section (3.9), the authors present long list of genes and speculated their role in cold resistance rather than reporting a solid finding.

Reply: The gene obtained by the transcriptome that "plays an important role in potato cold resistance" is just a very valuable "candidate gene." It is a "scientific hypothesis" that requires subsequent experiments such as gene function verification to verify.。Therefore, the results obtained by the transcriptome are “suggest” or “might”, rather than exact results such as “indicate”.

  1. Section 3.11: The section is difficult to follow because it contains a list of metabolite names and speculative statement without any clear flow. For example in the following sentence: " In this study, “amino acid metabolism” mainly include “arginine and proline metabolism”, “Phenylalanine metabolism”, “tryptophan metabolism”.

Reply: The text and the picture already correspond  Figure 11

L-Cystine might both related to direct cold resistance of potato and involved in regulating potato cold resistance". Long list of metabolism is mentioned and speculated by stating "might both related to direct cold resistance of potato and involved in regulating potato cold resistance ". The word "both" refers to two items but here you mentioned long lists and try to close up with both, it is not logical.

Reply: Based on the criteria for screening “metabolites might linked to potato cold resistance” in this study to selecte metabolites. line 144   

Based on the criteria for screening “gene might linked to potato cold resistance” in this study to selecte gene. line 154-158

Direct cold resistance and indirect cold resistance are only definitions of metabolites and genes, but metabolism pathways are not described in this way.

Discussion:

In most of the cases the discussion section is a repetition of the results section rather than interpret and explain the results. This section mainly relies on external citations to explain their findings but the discussion section should primarily interpret the result using citations to support their interpretations. The authors use inconsistent usage of both text and number citation format (eg., Line 506 and 610). The majority of this section is not properly organized and contains somehow disjointed facts rather than a cohesive scientific argument. In general the discussion section needs a thorough re-writing by focusing on explaining the biological significance of the findings rather than simply listing data and external citations.

Reply: corrected

Comments on the Quality of English Language

The text is poorly written; it requires professional English language editing. 

Reply: Thank you for this suggestion. We highly value the importance of professional language polishing. In our current revision, we have focused primarily on addressing the substantial scientific points raised by all reviewers. Our plan is to perform a comprehensive language editing and proofreading of the entire manuscript once all the scientific revisions are finalized to ensure consistency and high quality. We will certainly do this before the final submission.

Reviewer 2 Report

Comments and Suggestions for Authors

Line 135: The "12–24h recovery" range is too wide. Why is it so variable? Why wasn't it constant?

Lines 143–145: Different measurement times are specified, but it is not clear which parameter was measured at which time.

Line 190: “after 7 days of 7-day cold stress” ; unnecessary repetition

Fig. 2: The language used in figure captions is very non-technical ("first line of pictures", "the picture from the left...") – not suitable for academic presentation.

Line 226: DAM numbers are given but statistical significance (FDR?) is not specified.

Lines 259–269: DEG numbers are given, but the biological interpretation is missing. Which gene families are dominant, which pathways are activated?

Lines 285–296: Only TF numbers are given, but the contribution of specific TF families (e.g., WRKY, NAC) is not commented.

Lines 333–345: KEGG paths are listed, but the reasons for their importance are not explained. Only a list is presented.

Line 497: “Atlantic showed significantly higher…” ; KY130 and KY140 were introduced, and the name “Atlantic” is used here for the first time. Could this be a typo? Can an explanation be provided?

Lines 555–577: TF examples are well presented, but the connection between these literature examples and this study is lacking. For example, a comparison such as "In contrast to our data..." could contribute to a clearer understanding of the relationship.

Line 624: “we employed analysis of physiological, metabolic, and transcriptome…”; It should read "we conducted physiological, metabolomic and transcriptomic analyses".

Line 636: “These findings will advance our knowledge…” should be expressed more cautiously (like “may contribute to…”).

In expressions such as "P < 0.05, P < 0.01", "P" should be lowercase and italicized (p).

Author Response

We thank the reviewer for their careful reading of our manuscript and for their insightful comments, which have helped us significantly improve the paper. We have revised the manuscript accordingly, and our point-by-point responses to the comments are provided below.

Line 135: The "12–24h recovery" range is too wide. Why is it so variable? Why wasn't it constant?

Reply: Potato tissue culture seedlings grow faster in suitable environments. Potatoes were subjected to 7 days of cold stress (-4℃) and recovered at 22-25℃ before phenotypic identification was carried out. Our laboratory has identified more than 30 varieties using this method. Some varieties can identify phenotypes after 12 hours of recovery, while some varieties have poor recovery effect after 12 hours. If the recovery time is long, new branches and leaves will grow out of the potato tissue culture seedlings, which will affect the accuracy of the results. Recovery results at 12h and 24h are almost the same.

Lines 143–145: Different measurement times are specified, but it is not clear which parameter was measured at which time.

Reply: 2.1.1 Experimental design for phenotyping with statistical evaluation    Line 100

-4℃ for 7 days

2.1.2 Measurement of caffeic acid and cinnamic acid levels using ELISA   Line 109

-4℃ for 12 h

2.1.3 Plant treatment and sampling for physiological assays   Line 113

-4°C for 6, 12, and 18 h

2.1.4 Sample preparation for metabolomic and transcriptomic analysis   Line 117

-4℃ for 12 h

Line 190: “after 7 days of 7-day cold stress” ; unnecessary repetition

Reply: corrected   Line181

Fig. 2: The language used in figure captions is very non-technical ("first line of pictures", "the picture from the left...") – not suitable for academic presentation.

Reply: corrected    Figure 2   Line 199

Line 226: DAM numbers are given but statistical significance (FDR?) is not specified.

Reply: The screening criteria for DAMs were established as follows: Variable importance in projection (VIP) ≥ 1 and fold change (FC) ≥ 2 or FC ≤ 0.5. DAMs refer to metabolites whose content changes significantly in different comparison groups. (Line 141-143).

DEGs were identified based on the following criteria: |log2FC| ≥ 1 and false discov-ery rate (FDR) < 0.05. (Line 153-154).

Lines 259–269: DEG numbers are given, but the biological interpretation is missing. Which gene families are dominant, which pathways are activated?

Reply: A macro description of the transcriptome data is given here. TF analysis line 281, KEGG analysis line 331, GO analysis line 304

Lines 285–296: Only TF numbers are given, but the contribution of specific TF families (e.g., WRKY, NAC) is not commented.

Reply: WRKY has been added, but the number of NACs is relatively small, so it has not been added line 281-295

Lines 333–345: KEGG paths are listed, but the reasons for their importance are not explained. Only a list is presented.

Reply: Figure 8 line 322   Figure A is part of the KY130 pathway, Figure B is all common KY140 pathways, and the text is a summary of all pathways  line 331

Line 497: “Atlantic showed significantly higher…” ; KY130 and KY140 were introduced, and the name “Atlantic” is used here for the first time. Could this be a typo? Can an explanation be provided?

Reply: corrected  line 503

Lines 555–577: TF examples are well presented, but the connection between these literature examples and this study is lacking. For example, a comparison such as "In contrast to our data..." could contribute to a clearer understanding of the relationship.

Reply: added  line 582-583

Line 624: “we employed analysis of physiological, metabolic, and transcriptome…”; It should read "we conducted physiological, metabolomic and transcriptomic analyses".

Reply: corrected  line 647

Line 636: “These findings will advance our knowledge…” should be expressed more cautiously (like “may contribute to…”).

Reply: corrected  line 660

In expressions such as "P < 0.05, P < 0.01", "P" should be lowercase and italicized (p).

Reply: corrected line 203

Reviewer 3 Report

Comments and Suggestions for Authors

The manuscript presents a useful metabolomic and transcriptomic analysis of cold-stress response in potato. The following comments are provided to help improve the manuscript.

Abstract: reword for clarity and properly identify abbreviations on first occurrence by writing out the word and placing its abbreviation behind it in parentheses.

Introduction: reword for clarity and flow.

Line 37: As an example of effective rewording, the sentence that begins on this line should read, “Although potato plants thrive in cool climates, they exhibit limited tolerance to freezing and low temperatures [2]. Cold stress is therefore one of the important environmental factors affecting the growth and development of potato species [3].

Line 48: A new paragraph should start here beginning with the sentence, “Omics-driven strategies are reshaping our understanding of cold adaptation in plants [4,12].”

Line 68: A new paragraph should start here and begin as follows: “Transcriptomics refers to the discipline that studies the transcription of genes in cells and the laws of transcription regulation at the RNA level. In combination with metabolomics, transcriptomic analysis serves as a powerful tool for deciphering plant responses to cold stress.”

Line 73: Rewording here could be similar to the following:

Recent transcriptomics reports have  provided genome-wide dissection of cold-responsive regulatory networks and molecular adaptations [12,18]. In two maize (Zea mays L.) inbred lines - M54 (chilling tolerant) and 753F (chilling-sensitive), cold stress treatments of 4 or 24 h revealed that the majority of cold-responsive genes were associated with photosynthesis, secondary metabolism, and signal transduction [19]. Transcriptomic analyses of cold-tolerant (NH5) and cold-sensitive (FH18) peanut (Arachis hypogaea L.) varieties indicated that lipid and fatty acid metabolism may also play a significant role in cold tolerance, with 59 identified cold-responsive genes involved in these metabolic pathways [20]. We recently elucidated the molecular mechanisms of cold stress response in rapeseed (Brassica napus L.) using transcriptomic analysis [21]. Profiling in the C18 (cold-tolerant) and C6 (cold-sensitive) rapeseed varieties demonstrated the involvement of genes associated with photosynthesis, redox dynamics, and energy metabolism in four-leaf seedlings under cold-stress [21].

            Line 95: “integrated” is misspelled.

            Line 108: This paragraph must be reworded to remove summarized results and to directly compare cinnamic acid and caffeic acid with respect to cold stress response and regulatory roles observed in other literature reports.

Materials and Methods: reword for clarity.

            Line 133: Define “MS medium”, reword or define “25001x with 16h per day”, and add container size here somewhere.

            Line 134: Reword to clearly define the experimental conditions (cold-stress versus cool or normal temperatures).

             Line 136 Remove results to read, “Leaves with >50% wilted surface area were considered damaged.”

            Line 139: Remove results sentence that begins “Both KY130…”

            Line 143: Perhaps describe treatments as being performed in triplicate here.

            Line 148: “quantitative” should not be capitalized.

            Line 158: Library construction and RNA sequencing were conducted by MetWare Biotechnology (Wuhan, China) using the MetWare database (MWDB) (MetWare Biotechnology Co., Ltd., Wuhan, China) for metabolite identification.

            Line 178: Define the 2-DDCt method or give a reference that does.

Results: reword for clarity, improve figure legends to avoid reader confusion.

            Line 187: Rewrite this paragraph to avoid restating the introduction directly and to begin a stepwise progression through the data, referring to Figure 1.

            FIGURE 1: Perhaps there should be a ruler along the vertical instead of the 1cm legend to make comparison clearer.

            FIGURE 2: Separate the top two graphs to their own figure, and completely redesign the rest of the figures graphs to emphasize select comparisons and completely describe them in the legend.

            FIGURES 3&4: Perhaps pair these down to one impactful set of visual comparisons and present other comparisons as a summary in the supplemental materials.

            FIGURES 5-8: Redesign with meaningful subtitles in the figure and only the most relevant datapoints included.

            FIGURES 9-12: Perhaps separate the part C from each, place as a combined figure in the Supplemental, and clarify their labels.

Discussion and Conclusion: capitalize “Conclusion”, reword for clarity, add more specific references to the Figures.

References: remove the extra set of numbers.

Comments on the Quality of English Language

The English language and data presentation obscure the findings and render the manuscript difficult to read and understand. Rewording and clarification throughout is advisable. 

Author Response

We thank the reviewer for these helpful comments. Our manuscript has been revised to address all concerns, and responses are provided below.

The manuscript presents a useful metabolomic and transcriptomic analysis of cold-stress response in potato. The following comments are provided to help improve the manuscript.

Abstract: reword for clarity and properly identify abbreviations on first occurrence by writing out the word and placing its abbreviation behind it in parentheses.

Reply: corrected

Introduction: reword for clarity and flow.

Line 37: As an example of effective rewording, the sentence that begins on this line should read, “Although potato plants thrive in cool climates, they exhibit limited tolerance to freezing and low temperatures [2]. Cold stress is therefore one of the important environmental factors affecting the growth and development of potato species [3].

Reply: corrected  line 40-43

Line 48: A new paragraph should start here beginning with the sentence, “Omics-driven strategies are reshaping our understanding of cold adaptation in plants [4,12].”

Reply: corrected  line 61-62

Line 68: A new paragraph should start here and begin as follows: “Transcriptomics refers to the discipline that studies the transcription of genes in cells and the laws of transcription regulation at the RNA level. In combination with metabolomics, transcriptomic analysis serves as a powerful tool for deciphering plant responses to cold stress.”

Reply: This suggestion is very good, thank you, due to Reviewer 1 's suggestion, I deleted this part

Line 73: Rewording here could be similar to the following:

Reply: I am grateful to you for helping me correct so many sentences. Due to the suggestion of Reviewer 1, I deleted this part

Recent transcriptomics reports have  provided genome-wide dissection of cold-responsive regulatory networks and molecular adaptations [12,18]. In two maize (Zea mays L.) inbred lines - M54 (chilling tolerant) and 753F (chilling-sensitive), cold stress treatments of 4 or 24 h revealed that the majority of cold-responsive genes were associated with photosynthesis, secondary metabolism, and signal transduction [19]. Transcriptomic analyses of cold-tolerant (NH5) and cold-sensitive (FH18) peanut (Arachis hypogaea L.) varieties indicated that lipid and fatty acid metabolism may also play a significant role in cold tolerance, with 59 identified cold-responsive genes involved in these metabolic pathways [20]. We recently elucidated the molecular mechanisms of cold stress response in rapeseed (Brassica napus L.) using transcriptomic analysis [21]. Profiling in the C18 (cold-tolerant) and C6 (cold-sensitive) rapeseed varieties demonstrated the involvement of genes associated with photosynthesis, redox dynamics, and energy metabolism in four-leaf seedlings under cold-stress [21].

            Line 95: “integrated” is misspelled.

Reply: This suggestion is very good, thank you, due to Reviewer 1 's suggestion, I deleted this part

            Line 108: This paragraph must be reworded to remove summarized results and to directly compare cinnamic acid and caffeic acid with respect to cold stress response and regulatory roles observed in other literature reports.

Reply: corrected  line 79-92

Materials and Methods: reword for clarity.

            Line 133: Define “MS medium”, reword or define “25001x with 16h per day”, and add container size here somewhere.

Reply: MS medium (Carrageenan) (Shijiazhuang Nutrition Tissue Culture Technology Co., Ltd., Shijiazhuang, China) Line 96-97

Tissue culture room maintained 2500lx with 16h per day. Low temperature incubator (Shanghai YiHeng Scientific Instruments Co., Ltd.) Line 97-99

            Line 134: Reword to clearly define the experimental conditions (cold-stress versus cool or normal temperatures).

Reply: 2.1.1 Experimental design for phenotyping with statistical evaluation    Line 100

-4℃ for 7 days

2.1.2 Measurement of caffeic acid and cinnamic acid levels using ELISA   Line 109

-4℃ for 12 h

2.1.3 Plant treatment and sampling for physiological assays   Line 113

-4°C for 6, 12, and 18 h

2.1.4 Sample preparation for metabolomic and transcriptomic analysis   Line 117

-4℃ for 12 h

             Line 136 Remove results to read, “Leaves with >50% wilted surface area were considered damaged.”

Reply: This sentence is not the result, but the definition of blade damage  line103-104

            Line 139: Remove results sentence that begins “Both KY130…”

Reply: The meaning of this sentence is that every individual of each material will be damaged, but the degree of injury will vary. There are no particularly extreme situations, such as one plant being not injured at all, and the leaves of the other plant are dead  line 106-107

            Line 143: Perhaps describe treatments as being performed in triplicate here.

Reply: corrected   line111-112

            Line 148: “quantitative” should not be capitalized.

Reply: corrected   line120

            Line 158: Library construction and RNA sequencing were conducted by MetWare Biotechnology (Wuhan, China) using the MetWare database (MWDB) (MetWare Biotechnology Co., Ltd., Wuhan, China) for metabolite identification.

Reply: Thank you for your suggestion. Since Reviewer 1 asked me to add sample collection and experimental methods for omics, there are more contents, I wrote about metabolism and transcription separately. line130-158

            Line 178: Define the 2-DDCt method or give a reference that does.

Reply: added    line 167-168

Results: reword for clarity, improve figure legends to avoid reader confusion.

Reply: corrected

            Line 187: Rewrite this paragraph to avoid restating the introduction directly and to begin a stepwise progression through the data, referring to Figure 1.

Reply: corrected   line180-181

            FIGURE 1: Perhaps there should be a ruler along the vertical instead of the 1cm legend to make comparison clearer.

Reply: Your suggestion is better, I refer to Cai et al. (2022)

Zhaoqin Cai, Zhipeng Cai, Jingli Huang, Aiqin Wang, Aaron Ntambiyukuri, Bimei Chen, Ganghui Zheng, Huifeng Li, Yongmei Huang, Jie Zhan Dong Xiao and Longfei He. Transcriptomic analysis of tuberous root in two sweet potato varieties reveals the important genes and regulatory pathways in tuberous root development   BMC Genomics   https://doi.org/10.1186/s12864-022-08670-x

            FIGURE 2: Separate the top two graphs to their own figure, and completely redesign the rest of the figures graphs to emphasize select comparisons and completely describe them in the legend.

Reply: corrected   line199

            FIGURES 3&4: Perhaps pair these down to one impactful set of visual comparisons and present other comparisons as a summary in the supplemental materials.

Reply: Your suggestion is better, I refer to Wang et al. (2021)

Wang X, Liu Y, Han Z, Chen Y, Huai D, Kang Y, Wang Z, Yan L, Jiang H, Lei Y and Liao B (2021) Integrated Transcriptomics and Metabolomics Analysis Reveal Key Metabolism Pathways Contributing to Cold Tolerance in Peanut. Front. Plant Sci. 12:752474.

 doi: 10.3389/fpls.2021.752474

            FIGURES 5-8: Redesign with meaningful subtitles in the figure and only the most relevant datapoints included.

Reply: Your suggestion is better, I refer to Lv et al. (2022)

Liangjie Lv, Ce Dong, Yuping Liu, Aiju Zhao, Yelun Zhang, Hui Li and Xiyong Chen. Transcription-associated metabolomic profiling reveals the critical role of frost tolerance in wheat BMC Plant Biology.  https://doi.org/10.1186/s12870-022-03718-2

            FIGURES 9-12: Perhaps separate the part C from each, place as a combined figure in the Supplemental, and clarify their labels.

Reply: Your suggestion is better, I refer to Lv et al. (2022)

Liangjie Lv, Ce Dong, Yuping Liu, Aiju Zhao, Yelun Zhang, Hui Li and Xiyong Chen. Transcription-associated metabolomic profiling reveals the critical role of frost tolerance in wheat BMC Plant Biology.  https://doi.org/10.1186/s12870-022-03718-2

Discussion and Conclusion: capitalize “Conclusion”, reword for clarity, add more specific references to the Figures.

Reply: added   line 646-661

References: remove the extra set of numbers.

Reply: deleted   line 671-811

Comments on the Quality of English Language

The English language and data presentation obscure the findings and render the manuscript difficult to read and understand. Rewording and clarification throughout is advisable.

Reply: Thank you for this suggestion. We highly value the importance of professional language polishing. In our current revision, we have focused primarily on addressing the substantial scientific points raised by all reviewers. Our plan is to perform a comprehensive language editing and proofreading of the entire manuscript once all the scientific revisions are finalized to ensure consistency and high quality. We will certainly do this before the final submission.

Round 2

Reviewer 1 Report

Comments and Suggestions for Authors

A professional English Language editor is still required

Comments on the Quality of English Language

A professional English Language editor is still required

Author Response

We thank the reviewer for this important point.

We highly value the importance of professional language polishing. In our current revision, based on your first review, we will thoroughly rewrite the discussion and resolve the remaining scientific issues raised by commentator 3. Our plan is to perform a comprehensive language editing and proofreading of the entire manuscript once all the scientific revisions are finalized to ensure consistency and high quality. We will certainly do this before the final submission.

Reviewer 2 Report

Comments and Suggestions for Authors

I have reviewed the revisions and clarifications made by the authors. I would like to express my gratitude.

Author Response

We thank the reviewer for their time and insightful comments.

We highly value the importance of professional language polishing. In our current revision, we will thoroughly rewrite the discussion section in response to the comments provided by Reviewer 1 and fully address all remaining scientific points raised by Reviewer 3. Our plan is to perform a comprehensive language editing and proofreading of the entire manuscript once all the scientific revisions are finalized to ensure consistency and high quality. We will certainly do this before the final submission.

Reviewer 3 Report

Comments and Suggestions for Authors

General: Rewording throughout the manuscript for clarity is still advisable. In the Results section particularly there are dropped articles and prepositions, cumbersome lists without punctuation, and quotations which do not appear to be meaningful. 

Introduction:

Line 39: remove "Chinese people's"

Line 111: insert the sentence beginning on line 124 directly in front of the sentence beginning with "All treatments".

Line 111: replace "as being" with "were".

Line 123: The subtitle should read Physiological indicators analysis

Line 124: This sentence should moved to line 111 as described above.

Line 129: add "according to the manufacturer's instructions" at the end of the sentence.

Results:

Line 252: this paragraph is cumbersome and the data should be better summarized.

Line 346: not sure what the quotation marks are for and the first sentence seems to be truncated at the start without capitalization.

Figures 2-12: The authors correctly referenced three papers, Wang, Cai, and Lv, in explaining the complexity of their figures and the legends are somewhat improved in line with those references. However, significant clarification and reworking of the language that describes and walks the reader through the figures in the Results section is still necessary. Also, the cited references feature figures with color-coding and arrangements that visually guide the reader through them--this is an advisable modification for final presentation of this manuscript.

Comments on the Quality of English Language

While it appears that changes have been made throughout, the language and grammar are still rendering the work difficult to read. 

Author Response

Thank you for this insightful comment. We agree that addressing this point has significantly improved the manuscript. We have revised the text accordingly as detailed below.

General: Rewording throughout the manuscript for clarity is still advisable.

In the Results section particularly there are dropped articles and prepositions, cumbersome lists without punctuation, and quotations which do not appear to be meaningful.

Reply: Thank you for this suggestion. We highly value the importance of professional language polishing. In our current revision, we have focused primarily on addressing the substantial scientific points raised by all reviewers. Our plan is to perform a comprehensive language editing and proofreading of the entire manuscript once all the scientific revisions are finalized to ensure consistency and high quality. We will certainly do this before the final submission.

Introduction:

Line 39: remove "Chinese people's"

Reply: corrected   line 41

Line 111: insert the sentence beginning on line 124 directly in front of the sentence beginning with "All treatments".

Reply: corrected 113-115

Line 111: replace "as being" with "were".

Reply: corrected line 116

Line 123: The subtitle should read Physiological indicators analysis

Reply: corrected  line 127

Line 124: This sentence should moved to line 111 as described above.

Reply: corrected  line 113-115

Line 129: add "according to the manufacturer's instructions" at the end of the sentence.

Reply: added   line 131

Results:

Line 252: this paragraph is cumbersome and the data should be better summarized.

Reply: corrected   line248

Line 346: not sure what the quotation marks are for and the first sentence seems to be truncated at the start without capitalization.

Reply: corrected 341-343

Figures 2-12: The authors correctly referenced three papers, Wang, Cai, and Lv, in explaining the complexity of their figures and the legends are somewhat improved in line with those references.

However, significant clarification and reworking of the language that describes and walks the reader through the figures in the Results section is still necessary.

Also, the cited references feature figures with color-coding and arrangements that visually guide the reader through them--this is an advisable modification for final presentation of this manuscript.

Thank you for your suggestion. We included multiple figures to ensure the results were presented clearly and comprehensively, supporting the robustness of our conclusions.
